# Guided Star-Shaped Masked Diffusion

Viacheslav Meshchaninov [* 1 2]   Egor Shibaev [* 1]   Artem Makoian [1]   Ivan Klimov [1]   Nikita Balagansky [3]
Daniil Gavrilov [3]   Aibek Alanov [2 4]   Dmitry Vetrov [1]

## Abstract

The performance of pre-trained masked diffusion models is often constrained by their sampling procedure, which makes decisions irreversible and struggles in low-step generation regimes. We introduce a novel sampling algorithm that works with pre-trained models and, after a lightweight fine-tuning of a single layer, significantly improves sample quality and efficiency. Our method reformulates the generation process using a star-shaped paradigm, which inherently allows for error correction. To make this process effective, we augment it with a learnable remasking module that intelligently identifies and revises likely errors. This approach yields a substantial quality boost, particularly when using a small number of sampling steps. We extensively ablate key components of our approach and show its usability in different scenarios. In experiments on text, and code generation, our sampling algorithm outperforms or matches existing methods. Code is available at https://github.com/EgorShibaev/G-Star.

## 1. Introduction

Diffusion probabilistic models have revolutionized continuous domains by leveraging iterative refinement, where progressive denoising allows early errors to be corrected in subsequent steps (Sohl-Dickstein et al., 2015; Song & Ermon, 2019; Ho et al., 2020). However, this self-correcting capability is largely absent in discrete masked diffusion (Gat et al., 2024; Lou et al., 2024; Sahoo et al., 2024). In these frameworks, the transition from [MASK] to a token is irreversible: once a prediction is made, the model commits to it permanently. This rigidity creates a critical bottleneck for

---
*Equal contribution  [1]Constructor University  [2]Higher School of Economics  [3]T-Tech  [4]FusionBrain Lab. Correspondence to: Viacheslav Meshchaninov <vmeshchani@constructor.university>.

*Proceedings of the 43$^{rd}$ International Conference on Machine Learning*, Seoul, South Korea. PMLR 306, 2026. Copyright 2026 by the author(s).

parallel generation, as the inconsistencies inevitably introduced by predicting multiple tokens simultaneously cannot be rectified without a mechanism for selective correction.

To address this limitation, several approaches have tackled the problem from different perspectives. For instance, Peng et al. (2025) and Zheng et al. (2023) utilize the diffusion model's own confidence in already generated tokens to determine which should be remasked. Others frame generation explicitly as an iterative refinement task: Liu et al. (2024) employ a two-stage planning-and-denoizing mechanism, while Song et al. (2025) finetune the model to recover sequences corrupted by bounded Levenshtein noise. Extending this idea, von Rütte et al. (2025) combines masked diffusion with a uniform diffusion process to enable token refinement. More recently, Wang et al. (2025) introduced a simpler yet effective strategy: randomly remasking a fraction of generated tokens during sampling.

Despite the variety of approaches highlighting the critical importance of error correction, a standard solution remains absent. The primary bottleneck stems from three fundamental limitations in existing techniques. First, methods relying on stochastic remasking are computationally inefficient, as they depend on chance to cover errors, necessitating numerous refinement steps. Second, confidence-based heuristics are inherently unreliable because discrete diffusion models are trained to optimize loss only on masked tokens, leaving their confidence on clean (unmasked) tokens uncalibrated. Third, methods that train a separate refiner often rely on uniform noise (replacing tokens with random entries from the vocabulary). This constitutes a significantly easier detection task than identifying the subtle, contextually plausible errors made by the diffusion model, leading to poor generalization during actual sampling.

To address the limitations of current remasking strategies, we propose a Guided Star-Shaped sampler for masked diffusion. A fundamental distinction between our approach and standard MDLM sampling (Sahoo et al., 2024) lies in the transition from state $\mathbf{x}_t$ to $\mathbf{x}_s$. In MDLM, clean tokens in $\mathbf{x}_t$ are strictly copied to $\mathbf{x}_s$, with only masked tokens being revealed. In contrast, our sampler first predicts a fully denoised estimate $\hat{\mathbf{x}}_0 \sim p_\theta(\cdot \mid \mathbf{x}_t)$, and then samples $\mathbf{x}_s \sim q(\cdot \mid \hat{\mathbf{x}}_0)$ independently of $\mathbf{x}_t$. While this decou-

pling offers significant flexibility for error correction, it inherently introduces stochasticity: since $\mathbf{x}_s$ is conditionally independent of $\mathbf{x}_t$ given the prediction $\hat{\mathbf{x}}_0$, the generation process risks losing context or drifting into incoherent text, as demonstrated in Section 4.2.

We propose to resolve this stability-flexibility trade-off through the following contributions:

- We theoretically and empirically demonstrate that the star-shaped sampler is compatible with standard pre-trained masked diffusion models, allowing it to be seamlessly integrated with classical MDLM sampling.

- We identify the optimal sampling phase for refinement, showing that remasking is ineffective during early generation stages due to insufficient context.

- We propose a learned error detector $g_\phi$ to govern the transition $\hat{\mathbf{x}}_0 \to \mathbf{x}_s$. Instead of training on uniform noise (substituting tokens with random vocabulary entries), $g_\phi$ is trained to explicitly detect and target errors made by the diffusion model. This targeted approach prevents unnecessarily remasking of correct tokens and significantly accelerates generation.

- We show that $g_\phi$ can be implemented as a lightweight head over the frozen diffusion model's activations, introducing negligible computational and memory overhead.

- We demonstrate the general applicability of our approach by reporting superior results on both natural language and code generation, and validating its effectiveness at scale.

## 2. Preliminaries

**Masked Diffusion Models.** We consider discrete tokens represented as one-hot vectors $\mathbf{x} \in \{0, 1\}^{|V|}$, where $|V|$ is the vocabulary size. A special `[MASK]` token is denoted by $\mathbf{m}$. The forward process corrupts an input $\mathbf{x}_0$ by progressively masking tokens over $T$ timesteps according to a noise schedule $\alpha_t$. The marginal distribution of the noisy state $\mathbf{x}_t$ is given by:

$$q(\mathbf{x}_t \mid \mathbf{x}_0) = \text{Cat}(\mathbf{x}_t; \alpha_t \mathbf{x}_0 + (1 - \alpha_t)\mathbf{m}). \quad (1)$$

The reverse process is parameterized by a neural network, $f_\theta(\mathbf{x}_t, t)$, which is trained to predict the probability distribution over the original data, $p_\theta(\mathbf{x}_0 \mid \mathbf{x}_t)$. This predicted distribution then conditions the analytical posterior:

$$q(\mathbf{x}_{t-1} \mid \mathbf{x}_t, \mathbf{x}_0) =$$
$$\begin{cases} \delta_{\mathbf{x}_t}(\mathbf{x}_{t-1}), & \text{if } \mathbf{x}_t \neq \mathbf{m}, \\ \text{Cat}\left(\mathbf{x}_{t-1}; \dfrac{(1 - \alpha_{t-1})\mathbf{m} + (\alpha_{t-1} - \alpha_t)\mathbf{x}_0}{1 - \alpha_t}\right), & \text{if } \mathbf{x}_t = \mathbf{m}. \end{cases}$$
$$(2)$$

The first case of this posterior, where an unmasked token is deterministically preserved ($\delta_{\mathbf{x}_t}$), reveals the model's core limitation: once a token is generated, it is frozen, making iterative error correction impossible. The network $f_\theta$ is typically trained by minimizing a weighted cross-entropy loss to predict $\mathbf{x}_0$ from $\mathbf{x}_t$:

$$\mathbb{E}_{t,\mathbf{x}_0,\mathbf{x}_t}\left[w'_t \sum_{i=1}^{L} \mathbf{1}\left(x_t^i = [\text{M}]\right) \log p_\theta^i\left(x_0^i \mid \mathbf{x}_t\right)\right], \quad (3)$$

where $L$ is a sequence length, $\mathbf{1}(\cdot)$ is an indicator function active **only on masked tokens**, and $w'_t$ is the weighting schedule.

**Remasking Diffusion Models.** Several methods have been proposed to revise generated tokens. One common strategy is stochastic unguided remasking, where the model randomly revisits parts of the sequence (Campbell et al., 2022; Wang et al., 2025). While easy to implement, this method is inefficient. Since it masks tokens blindly, it often removes correct words, forcing the model to take many extra steps to fix the actual errors.

The second category attempts to solve this by directly modifying the diffusion process (von Rütte et al., 2025; Havasi et al., 2025; Liu et al., 2024). While these methods introduce new training objective to support iterative refinement, they suffer from two major downsides. First, they necessitate retraining the entire heavy diffusion backbone, which could be computationally prohibitive. Second, they are typically trained to correct uniform or Levenstein noise. This constitutes a significantly easier task than identifying the subtle, contextually plausible errors produced by the model during actual generation, limiting their effectiveness.

Finally, confidence-based methods try to avoid retraining by using the model's own confidence scores to find errors (Peng et al., 2025; Zheng et al., 2023). However, in masked diffusion the training objective (see Equation (3)) supervises only masked positions, while unmasked tokens are typically copied from $x_t$ and incur no direct loss. As a result, the model's probability estimates on already-generated (unmasked) tokens are not explicitly calibrated to reflect correctness, making confidence-based error detection unreliable in practice.

**Concurrent Work.** At the time of writing, two concurrent studies also explore learned refinement. Kim et al. (2025) shares the same motivation but propose fine-tuning the diffusion backbone alongside a remasking model. Huang et al. (2025) introduce a confidence-prediction module with four transformer layers, trained to detect random word replacements. Unlike these approaches, we design our method as a universal plug-and-play module. It allows for equipping any off-the-shelf masked diffusion model with a lightweight, learnable error corrector, thereby significantly boosting gen-

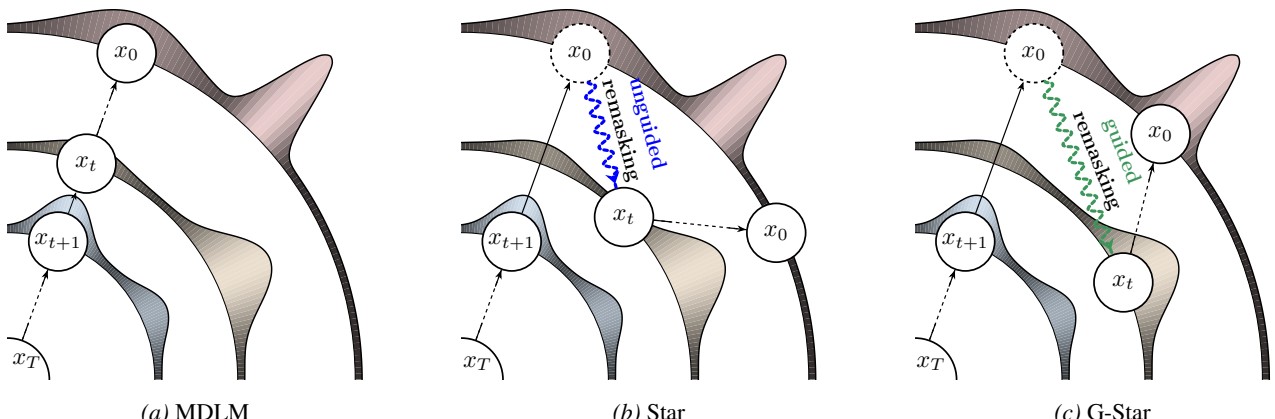

*(a)* MDLM        *(b)* Star        *(c)* G-Star

*Figure 1.* **Pipeline comparison of MDLM, Star, and G-Star sampling.** Each panel illustrates how a sampler moves from a noisy state $\mathbf{x}_T$ toward a clean sample $\mathbf{x}_0$. **(a) MDLM** follows an irreversible trajectory: once a token is unmasked, it is copied forward and cannot be revised. **(b) Star** routes each transition through a fully denoised draft $\hat{\mathbf{x}}_0$ and then randomly remasks tokens, enabling revision but wasting refinement steps on already correct tokens. **(c) G-Star** uses the same star-shaped transition, but replaces random remasking with an error predictor $g_\phi$ that selectively remasks likely incorrect tokens. This targeted remasking focuses refinement on probable errors while preserving the stability of already plausible context.

eration speed and quality while keeping the computational overhead minimal.

**Star-Shaped Diffusion Paradigm.** Okhotin et al. (2023) introduced the Star-Shaped paradigm, a process where states $\mathbf{x}_t$ and $\mathbf{x}_s$ are conditionally independent given $\mathbf{x}_0$. While theoretically appealing, this independence injects substantial stochasticity, causing consecutive states to diverge and potentially leading the generation into low-quality modes. To counteract this instability, the original authors relied on calculating sufficient statistics of the tail distribution that mandates a unique noise schedule and full model retraining.

While inspired by the general intuition of Okhotin et al. (2023), we adopt the Star-Shaped paradigm to the domain of Masked Diffusion, adapting its core principles while fundamentally altering the execution mechanism to ensure practicality. We identify that the inherent stochasticity of the process is disruptive in early generation stages, thus, we restrict its use to the end of the trajectory, where it serves as an effective refinement mechanism. Crucially, distinct from Okhotin et al. (2023), we derive a formulation that allows this sampler to be applied to off-the-shelf masked diffusion models (e.g., MDLM). This eliminates the need for expensive retraining, a capability absent in the original framework.

# 3. Guided Star-Shaped Masked Diffusion

In this section, we introduce a star-shaped paradigm for masked diffusion and then present a learned error predictor that makes the sampling process efficient and targeted. Figure 1 summarizes the difference between standard MDLM sampling, unguided star-shaped refinement, and our guided G-Star sampler.

## 3.1. Star-Shaped Masked Diffusion

We redefine the joint distribution of the forward process. Instead of conditioning each latent state $\mathbf{x}_t$ on its immediate predecessor $\mathbf{x}_{t-1}$, we make all latent states conditionally independent given the original data $\mathbf{x}_0$:

$$q(\mathbf{x}_{1:T} \mid \mathbf{x}_0) = \prod_{t=1}^{T} q(\mathbf{x}_t \mid \mathbf{x}_0). \qquad (4)$$

Here, each $q(\mathbf{x}_t \mid \mathbf{x}_0)$ is the same marginal distribution defined in Equation 1. This "star-shaped" structure, where all latents connect directly to $\mathbf{x}_0$, fundamentally alters the process dynamics. It explicitly permits non-monotonic transitions; for instance, a token can be masked at a timestep $s$ and become unmasked at a later timestep $t > s$, a scenario forbidden in the standard Markovian chain for masked diffusion (Sahoo et al., 2024).

This change simplifies the reverse posterior: $q(\mathbf{x}_{t-1} \mid \mathbf{x}_t, \mathbf{x}_0) = q(\mathbf{x}_{t-1} \mid \mathbf{x}_0)$. Following the standard diffusion paradigm, we construct the generative transition $p_\theta(\mathbf{x}_{t-1} \mid \mathbf{x}_t)$ by first predicting an estimate of the clean data, $\hat{\mathbf{x}}_0 \sim \text{Cat}(\cdot, f_\theta(\mathbf{x}_t, t))$, and then sampling from the corresponding posterior:

$$p_\theta(\mathbf{x}_{t-1} \mid \mathbf{x}_t) = q(\mathbf{x}_{t-1} \mid \mathbf{x}_0 = \hat{\mathbf{x}}_0). \qquad (5)$$

Crucially, note that this formulation differs from the standard MDLM posterior in Equation (2) by the **absence of conditioning on** $\mathbf{x}_t$.

Intuitively, each step of the generative process involves two stages: (1) the model examines the current state $\mathbf{x}_t$ and forms a complete hypothesis about the final, clean data $\hat{\mathbf{x}}_0$; (2) it then generates the next, less noisy state $\mathbf{x}_{t-1}$ by

---

**Algorithm 1** Training the Error Predictor $g_\phi$

1: **Input:** Dataset $\mathcal{D}$, pre-trained diffusion model $f_\theta$, learning rate $\eta$, denoiser temperature $\tau_{\text{denoiser}}$.
2: **Output:** Trained error predictor $g_\phi$
3: **while** not converged **do**
4:      Sample batch $\{\mathbf{x}_0\} \sim \mathcal{D}$
5:      ▷ Simulate denoising and identify errors $y$
6:      $t \sim \mathcal{U}(0, 1)$
7:      $\mathbf{x}_t \sim q(\cdot \mid \mathbf{x}_0)$
8:      $\hat{p}_0 \leftarrow \text{Softmax}(\frac{f_\theta(\mathbf{x}_t)}{\tau_{\text{denoiser}}})$
9:      $\hat{\mathbf{x}}_0 \sim \text{Cat}(\cdot; \hat{p}_0)$
10:      $y \in \{0, 1\}^L$, where $y_i = \mathbb{I}(\hat{\mathbf{x}}_{0,i} \neq \mathbf{x}_{0,i})$

11:      ▷ Train the error predictor
12:      $p \leftarrow \text{Softmax}(g_\phi(\hat{\mathbf{x}}_0))$
13:      $\mathcal{L}_\phi \leftarrow -\frac{1}{L} \sum_{i=1}^{L} \big[ y_i \log p_i$
                 $+ (1 - y_i) \log(1 - p_i)\big]$
14:      $\phi \leftarrow \phi - \eta \nabla_\phi \mathcal{L}_\phi$
15: **return** $g_\phi$

---

**Algorithm 2** Guided Sampling Step

1: **Input:** Current state $\mathbf{x}_t$, current time $t$, diffusion model $f_\theta$, error predictor $g_\phi$, denoiser temperature $\tau_{\text{denoiser}}$, nucleus probability $p_{\text{nucleus}}$, error predictor temperature $\tau_{\text{remask}}$
2: **Output:** Next state $\mathbf{x}_{t-1}$
3: ▷ Predict and sample a proposal clean state
4: $\hat{p}_0 \leftarrow \text{NucleusFilter}(\text{Softmax}(\frac{f_\theta(\mathbf{x}_t)}{\tau_{\text{denoiser}}}), p_{\text{nucleus}})$
5: $\hat{\mathbf{x}}_0 \sim \text{Cat}(\cdot; \hat{p}_0)$

6: ▷ Identify and select most likely errors
7: $\text{logits}_{\text{err}} \leftarrow g_\phi(\hat{\mathbf{x}}_0)$
8: $N \leftarrow \lceil (1 - \alpha_{t-1}) \cdot L \rceil$
9: $\mathcal{M} \leftarrow \text{SampleKNoRep}(\frac{\text{logits}_{\text{err}}}{\tau_{\text{remask}}}, N)$

10: ▷ Construct next state via targeted remasking
11: $\mathbf{x}_{t-1,i} \leftarrow \begin{cases} \mathbf{m}, & \text{if } i \in \mathcal{M} \\ \hat{\mathbf{x}}_{0,i}, & \text{otherwise} \end{cases}$
12: **return** $\mathbf{x}_{t-1}$

---

applying the forward noising process to this hypothesis, **independently of the current state** $\mathbf{x}_t$. This independence allows the model to revise its previous decisions, as it enables the **remasking of any previously generated token in** $\mathbf{x}_t$, regardless of its prior status (see Figure 1).

Notably, this star-shaped sampling process establishes a direct connection to the ReMDM framework (Wang et al., 2025). Specifically, our sampler is mathematically equivalent to the ReMDM sampler when its remasking parameter is set to $1 - \alpha_s$ (see proof in Appendix A.2).

A crucial consequence of this formulation is its compatibility with existing models. Consistent with (Wang et al., 2025), our variational lower bound (VLB) for this process can be simplified to a weighted cross-entropy objective, structurally identical to Equation (3).

**Claim 1.** *The VLB for the star-shaped process simplifies to the objective, which has the same functional form as the standard masked diffusion objective but with different timestep-dependent weights $w'_t$.* (Proof in Appendix A.1).

The structural similarity between our training objective and the standard MDLM loss motivates the reuse of pre-trained MDLM weights as an effective practical strategy. We empirically confirm this approach, finding that it allows our sampler to achieve strong performance without any fine-tuning.

### 3.2. Learned Error Correction

While the sampler described in Equation 5 enables error correction, it is inefficient. The remasking process is non-selective — it samples from $q(\mathbf{x}_{t-1} \mid \hat{\mathbf{x}}_0)$, which is just as likely to mask a correct token as an incorrect one. This neg-

atively impacts both sampling efficiency and final sample quality.

To rectify this, we introduce a secondary model: an error predictor $g_\phi$, which learns to identify which tokens the primary diffusion model $f_\theta$ is likely to get wrong. This allows us to focus the procedure on probable errors.

**Training the Error Predictor.** The purpose of the error predictor, $g_\phi$, is to identify tokens that the main diffusion model, $f_\theta$, is likely to generate incorrectly. To train it, we simulate this error-making process. First, we take a clean sequence $\mathbf{x}_0$ from the training data and apply the forward diffusion process to obtain a masked state $\mathbf{x}_t$. Next, we feed $\mathbf{x}_t$ to the frozen diffusion model $f_\theta$ and sample a discrete draft $\tilde{\mathbf{x}}_0 \sim \text{Cat}(\cdot, f_\theta(\mathbf{x}_t, t))$. The predictor is then trained with binary labels

$$e_i = \mathbb{I}[\tilde{x}_{0,i} \neq x_{0,i}], \tag{6}$$

which mark positions where the sampled draft disagrees with the paired clean sequence.

This target is a practical proxy for token-level error likelihood rather than a gold semantic-correctness label. In open-ended generation, a token can differ from the reference while still being valid. Nevertheless, this proxy is useful because it directly exposes the predictor to mistakes made by the frozen diffusion model under its own denoising distribution, without requiring external semantic annotation or retraining the backbone. In our experiments, alternative targets based on synthetic random corruptions or external language-model scores were less effective, while the disagreement target produced the strongest refinement frontier.

We also deliberately restrict the predictor's inputs. Apart from task conditioning when present, $g_\phi$ receives the sam-

pled draft $\tilde{\mathbf{x}}_0$ but not the noisy state $\mathbf{x}_t$, the timestep $t$, or the denoiser confidence scores. This prevents a conservative shortcut. Under the standard MDLM training distribution, unmasked positions in $\mathbf{x}_t$ are copied from the ground truth, so conditioning on $\mathbf{x}_t$ or confidence can teach the predictor that visible or high-confidence tokens should not be remasked. By removing these cues, the predictor must judge the draft itself and can learn to select tokens for actual correction rather than collapsing to a "do not revise" policy. The entire procedure is detailed in Algorithm 1.

**Inference with Targeted Remasking.** During generation, we incorporate the trained error predictor $g_\phi$ to guide the remasking process, replacing the sampler's indiscriminate selection of tokens with a targeted procedure (see Figure 1). The process for each sampling step from $\mathbf{x}_t$ to $\mathbf{x}_s$, detailed in Algorithm 2, proceeds as follows. First, the main diffusion model $f_\theta$ generates a clean data candidate, $\hat{\mathbf{x}}_0$. This candidate is then scored by the error predictor $g_\phi$ to obtain error logits for each token. These logits, scaled by a temperature $\tau_{\text{remask}}$, are then used to sample the $N$ locations without repetitions via Gumbel-Top-K trick sampling (Kool et al., 2019), where $N$ is determined by the noise schedule. The next state $\mathbf{x}_s$ is then formed by reverting these targeted tokens in $\hat{\mathbf{x}}_0$ back to the [MASK] symbol. By directing the model's capacity toward correcting probable errors, this targeted approach significantly boosts sampling efficiency and quality. As detailed in Sections 4–5, these gains are achieved with negligible parameter overhead (see Appendix D).

**Justification for the Star-Shaped Paradigm.** It is worth noting why the choice of the star-shaped paradigm introduced in Section 3.1 is essential for effectively training the reverse transition in masked diffusion. Alternatively, one could attempt to train a refinement model directly within the standard MDLM (Equation (2)) or ReMDM (Equation (18)) paradigms by conditioning on the current state $\mathbf{x}_t$ (i.e., learning $q(\mathbf{x}_{t-1} \mid \mathbf{x}_t, \hat{\mathbf{x}}_0)$). However, such an approach faces a critical training-inference mismatch. During training, the unmasked tokens in $\mathbf{x}_t$ are strictly derived from the ground truth $\mathbf{x}_0$. Consequently, a model conditioned on $\mathbf{x}_t$ learns to treat all observed tokens as immutable, effectively collapsing into a confidence estimator that rarely attempts to correct errors. While this issue could be partially mitigated by augmenting $\mathbf{x}_t$ with synthetic noise (e.g., random word replacement), such heuristics fail to approximate the complex, structural error distribution of the diffusion model (as we show in Section 4.4) and introduce additional distributional shift atop the intrinsic forward-backward diffusion mismatch. Our star-shaped formulation avoids this pitfall by structurally decoupling the generated state from the specific token choices in $\mathbf{x}_t$, thereby enabling error correction without reliance on synthetic perturbations.

## 4. Analysis

We validate our method's key components by analyzing: (1) the optimal generation phase for remasking; (2) the impact of guidance on quality and efficiency; (3) refinement performance within the loop protocol; and (4) the error predictor's architectural efficiency.

### 4.1. Experimental Setup

All analytical experiments are conducted on the OpenWebText (OWT) dataset (Gokaslan & Cohen, 2019), tokenized using the standard gpt-2 tokenizer (Radford et al., 2019). For these experiments, we fine-tuned the publicly available MDLM checkpoint from Sahoo et al. (2024) for unconditional generation of 128 and 512-token sequences, padding shorter outputs where necessary. We generate 5,000 samples for each configuration and assess performance using a suite of three complementary metrics. Sample quality and local coherence are measured via generative Perplexity (PPL), computed using a pre-trained GPT-2 LARGE model (Radford et al., 2019). Lexical variety is quantified by the Diversity (DIV) score, defined as $\text{div}(y) = \prod_{n=2}^{4} \frac{\#\text{ unique } n\text{-grams in } y}{\#\ n\text{-grams in } y}$. Finally, to provide a more holistic assessment that balances quality with diversity, we report the MAUVE score (Pillutla et al., 2021), which measures the distributional alignment between the generated and reference texts.

### 4.2. When to Use the Star-Shaped Sampler?

Our initial experiments revealed a critical insight: the pure star-shaped (Star) sampler, when applied across the entire generation trajectory, exhibits poor performance and often leads to degenerate text. This observation motivated our central hypothesis: the generation process is not monolithic but consists of two distinct phases, each benefiting from a different sampling strategy. We posit that the initial phase requires a stable, structure-building sampler, while the final phase benefits from an error-correcting one. To test this hypothesis, we first analyze this phenomenon in a simplified setting involving the generation of 128-token sequences from the OWT dataset.

**Phase 1: the Challenge of Early-Stage Generation.** In the early stages of generation (high $t$), a large fraction of tokens is masked. The star-shaped sampler's strategy of predicting a full $\hat{\mathbf{x}}_0$ requires the model to generate a large number of new tokens conditioned on a very sparse context. While these newly generated tokens may be individually plausible with respect to the unmasked context, they often lack mutual coherence among themselves. The problem is exacerbated by the subsequent step of the star-shaped process: the independent, random remasking of all tokens in this new hypothesis. This process may preserve a large

fraction of the newly generated, yet mutually incoherent, tokens while masking others that provided the original context. As a result, the input for the next iteration becomes an increasingly fragmented and incoherent context. This complicates the subsequent prediction task, causing errors to compound over iterations and ultimately leading to the observed text degradation.

This generative incoherence is empirically captured in Figure 2 (right), where the pure star-shaped (Star) sampler (light green line) demonstrates significantly lower step-to-step similarity than the standard MDLM (red line). This metric, defined as the fraction of matching tokens in the predicted clean data ($\hat{\mathbf{x}}_0$) between adjacent steps,. The low score for Star sampler confirms that its generation process struggles to build upon a coherent structure. This ultimately leads to text degradation, as reflected by its near-zero MAUVE score (see Figure 3 at $t_{on} = 1.0$). In contrast, the MDLM's incremental, token-by-token generation ensures high step-to-step similarity, allowing it to stably construct a coherent draft.

**Phase 2: the Power of Late-Stage Refinement.** While the MDLM's stability is advantageous for initial structure-building, its irreversible nature limits its ability to correct errors. This is where the star-shaped paradigm excels. In the late stages of generation (low $t$), the vast majority of tokens are already determined, providing a strong, coherent conditioning context. Remasking a small fraction of these tokens and repredicting them from a global perspective $\hat{\mathbf{x}}_0$ becomes a powerful mechanism for error correction, rather than a source of instability.

This effect is visible in Figure 2 (left). When our hybrid Star+ sampler switches from MDLM to star-shaped sampler at step 90 (dotted line), its perplexity (green line) begins to decrease more rapidly than the pure MDLM baseline, ultimately achieving a superior final score. This demonstrates that the Star sampler is highly effective at refining an already well-formed text.

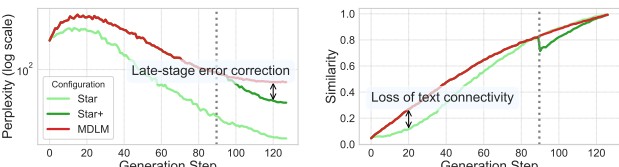

*Figure 2.* Analysis of the star-shaped (Star) sampler's dynamics. *(Left)* Perplexity and *(Right)* step-to-step similarity over the generation trajectory for three configurations: MDLM, Star, and our hybrid approach (Star+), which switches from MDLM to Star at step 90 (dotted line).

**Empirical Validation: Finding the Optimal Transition Point.** To validate this two-phase hypothesis and identify the optimal transition point, we conduct an ablation study on the activation time, $t_{on}$. The sampler operates as a standard

MDLM until time $t_{on}$, after which it switches to the star-shaped paradigm. Figure 3 plots the final MAUVE score as a function of $t_{on}$. The results provide strong empirical support for our hypothesis. Performance is poor for both pure samplers ($t_{on} = 1.0$ for pure Star and $t_{on} = 0.0$ for pure MDLM) but peaks at $t_{on} \approx 0.3$. This confirms that the most effective strategy is to leverage the MDLM process for the initial $60 - 80\%$ of the generation to build a coherent draft, and then activate the star-shaped sampler for the final $20 - 40\%$ for global refinement.

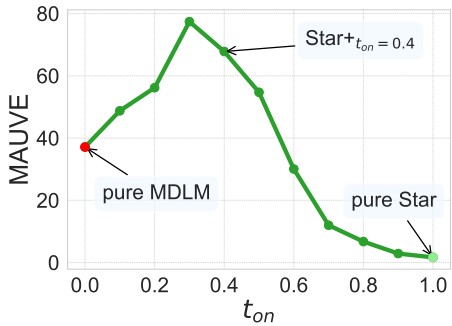

*Figure 3.* The impact of the star-shaped sampler's activation time ($t_{on}$) on generation quality. We plot the final MAUVE score for a hybrid sampler that switches from MDLM to Star at time $t_{on}$.

### 4.3. Guided Star-Shaped Sampler

The preceding analysis established that a hybrid sampler (Star+) effectively refines text in the late stages of generation. However, its reliance on unguided remasking is inherently sample-inefficient. This raises a central question: can we significantly improve performance by replacing this stochastic process with a targeted, intelligent one? In this section, we test this hypothesis by introducing our full proposed method, the **Guided Star-Shaped sampler (G-Star)**, which uses an error predictor to focus the refinement process exclusively on likely errors. We posit that the primary advantage of this targeted approach will manifest in computationally constrained, few-step generation regimes, where the efficiency of each correction step is paramount.

To validate this, we perform a direct comparison between the unguided Star+ and our guided G-Star+ (both employ the identical hybrid switching schedule, $t_{on} = 0.2$) sampler on the task of generating 512-token sequences from Open-WebText, evaluating across a range of sampling step counts from 32 to 512. The results are presented in Figure 4.

The empirical evidence strongly supports our hypothesis. The MAUVE scores (left panel) show that while both hybrid samplers outperform the MDLM baseline, the guided G-Star+ variant achieves significantly higher distributional fidelity. Crucially, the performance gap is most pronounced in the medium-step regimes of 64-256 steps. While at the

32-step mark all samplers struggle, our guided approach still shows a modest advantage. This gap then narrows as the step budget increases.

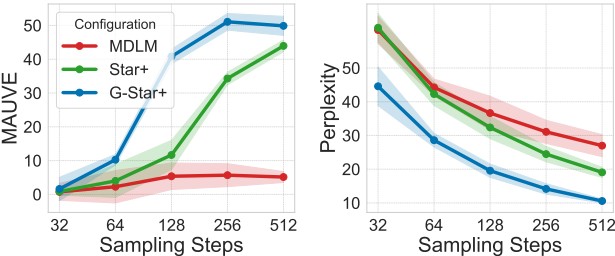

*Figure 4.* Performance comparison in few-step generation regimes. Guided sampler (G-Star+) consistently outperforms the unguided Star+.

This dynamic has a clear and intuitive explanation. With a large number of sampling steps, even an unguided remasking process has a high probability of eventually correcting most errors, causing the performance of the two samplers to converge. However, when each step is critical, the intelligent targeting provided by the error predictor becomes the deciding factor. By focusing the model's capacity on the most probable errors, the guidance mechanism ensures that each refinement step is maximally impactful. This enables the generation of higher-quality text with a significantly reduced computational budget, highlighting the practical advantage of our guided approach.

**Qualitative refinement trajectories.** We further visualize the mechanics of guided remasking in Figure 14. Starting from the same MDLM draft, unguided Star+ and G-Star+ quickly produce different refinement trajectories. Star+ remasks tokens in a scattered, largely indiscriminate pattern, including tokens that already form plausible context. In contrast, G-Star+ produces more structured remasking patterns, concentrating revisions on localized regions that are more likely to contain errors. This qualitative behavior supports the main mechanism of our method: the error predictor does not simply increase the amount of remasking, but changes where remasking is applied.

### 4.4. Iterative Refinement Regime

To further evaluate refinement capabilities of our sampler, we adopt the "loop schedule" from ReMDM (Wang et al., 2025). This protocol consists of three phases: (1) rapid drafting via standard MDLM; (2) iterative refinement at a constant noise level ($\alpha_t = 0.9$); and (3) final completion. Intuitively, this schedule accelerates the initial generation to reallocate the computational budget toward extended error correction (visualized in Figure 9).

We evaluate 512-token generation on OpenWebText, comparing against leading remasking strategies compatible with

off-the-shelf, frozen diffusion models: confidence-based samplers (P2-planner (Peng et al., 2025), RDM (Zheng et al., 2023)), refinement-based DDPD (Liu et al., 2024), and random remasking (ReMDM (Wang et al., 2025)). All baselines utilize the full 512-step budget, whereas our method is evaluated at 512, 256, and 128 steps. Crucially, while achieving strong results with ReMDM requires extensive, per-schedule tuning of the hyperparameter $\eta$ (see Appendix C.1), our unguided G-Star-loop sampler achieves competitive performance without such manual intervention, highlighting a key practical advantage of the star-shaped formulation.

Beyond scheduling, we observe that the denoiser's temperature ($\tau_{\text{denoiser}}$) provides a complementary axis of control over the generation process (see Appendix C.3). By adjusting the softmax temperature applied to the denoiser's output logits, the sampler can smoothly trade off between perplexity and diversity. Crucially, since the optimal temperature varies across different methods, relying on a single fixed value yields an incomplete comparison. Therefore, we vary $\tau_{\text{denoiser}}$ to construct Pareto frontiers, enabling us to evaluate each method across the entire quality-diversity spectrum rather than performing a limited point-wise comparison.

The Pareto fronts in Fig. 5a highlight the decisive advantage of our approach. Even with a $4\times$ reduction in step budget (128 vs. 512 steps), G-Star-loop attains superior perplexity and diversity compared to all full-budget baselines. This result validates the importance of targeted remasking: by guiding the sampler to focus solely on likely errors, we can afford a much faster initial drafting phase followed by precise, sparse corrections.

Among the baselines, we observe that simple random remasking (ReMDM) outperforms confidence-based sampling (P2-planner, RDM) and matches planners trained on uniform noise (DDPD). This performance gap stems from the remasking dynamics: while ReMDM ensures frequent and broad error coverage (see Appendix F), the uniform-noise planner is overly conservative (see Figure 12). Beyond this conservatism, DDPD suffers from a severe computational bottleneck. Specifically, it employs a guidance model, structurally equivalent to the diffusion backbone, at every step to plan the denoising trajectory, doubling both the memory footprint and inference latency. In contrast, our G-Star approach restricts remasking to a specific phase of the trajectory and, as we demonstrate in the next section, can be implemented via lightweight heads atop the frozen diffusion backbone, making the memory overhead negligible.

### 4.5. Error Predictor Capacity and Efficiency

**Impact of Predictor Capacity.** We investigate the trade-off between predictor size and refinement quality by comparing three configurations: a single fine-tuned block (1B, F), and full 12-block models with either full (12B, F) or head-

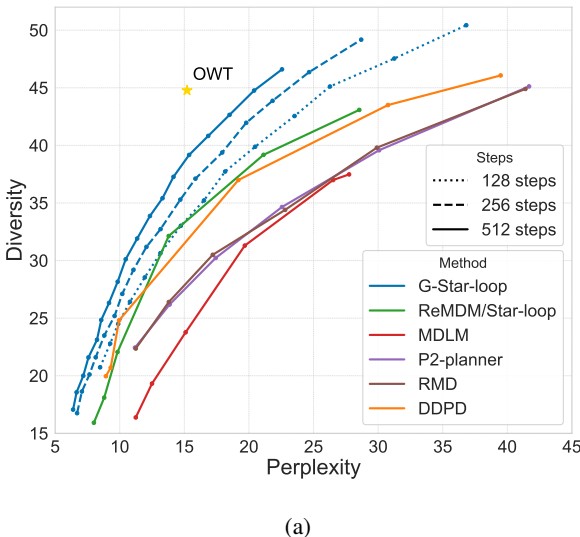

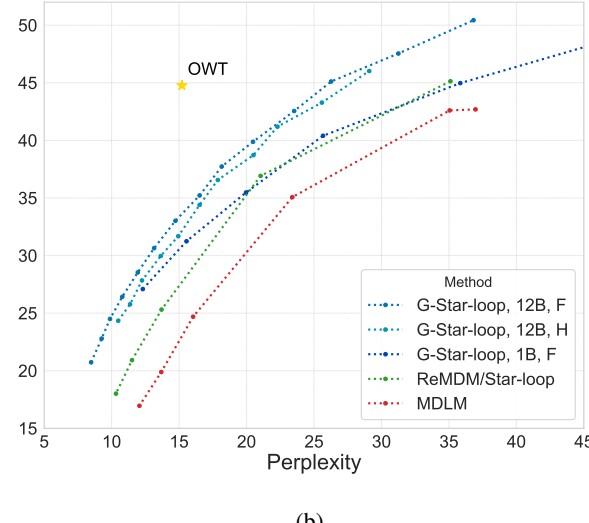

| (a) | (b) |

*Figure 5.* Pareto fronts for different methods obtained by varying the denoiser temperature. The left plot compares MDLM, Star/ReMDM-loop, P2-planner, RDM, DDPD and G-Star-loop, while the right plot compares different G-Star configurations (12B–F, 12B–H, 1B–F) against Star/ReMDM-loop and MDLM for 128 denoising steps.

only (`12B,H`) tuning. Evaluation in the challenging 128-step regime (Fig. 5b) reveals that the parameter-efficient `12B,H` variant closely tracks the fully fine-tuned `12B,F`, suggesting that pre-trained representations suffice for error detection. In contrast, the lightweight `1B,F` model is significantly less effective, offering marginal gains only at low diversity. Overall, this suggests that while head-only training offers an excellent efficiency–quality trade-off, more aggressive capacity reduction can noticeably degrade refinement performance.

**Robustness Across Domains.** We further assess whether the predictor generalizes beyond its training distribution. Evaluating the OWT-trained predictor on diverse unseen domains, including code, mathematics, daily news and stories, reveals remarkable stability (see Appendix C.4). The predictor maintains high classification performance (AUC-ROC $\geq 0.90$) across all shifts without fine-tuning. This suggests that the model successfully learns to identify intrinsic artifacts of the diffusion model.

**Computational Overhead.** Finally, we quantify deployment costs (full benchmarks in Appendix D). The robust `12B` configuration increases inference latency by approximately 50% due to the additional backbone passes required for targeted remasking. However, the memory footprint remains minimal: by employing the parameter-efficient `12B,H` strategy, we reuse the frozen backbone weights and introduce only a lightweight classification head, resulting in negligible parameter overhead. Ultimately, G-Star offers a highly favorable trade-off between computational cost and generation quality. As demonstrated in our Pareto analysis

(Fig. 5b), the 128-step G-Star sampler not only achieves a superior quality-diversity balance compared to the unguided 512-step ReMDM but also accelerates inference by nearly $3\times$ (3.43s vs. 9.16s).

# 5. Empirical Evaluation

Having analyzed the internal mechanics and key components of our sampler on the OWT dataset, we now turn to evaluating its performance and general applicability in a broader context. In this section, we benchmark G-Star on two challenging generative tasks: (1) **large-scale language modeling**, where we assess performance on downstream benchmarks to validate its effectiveness at scale, and (2) **source code generation** on the Conala benchmark (Yin et al., 2018).

## 5.1. Application to Large-Scale Instruction-Tuned Model

In this section we investigate whether our guided sampler can enhance the performance of an instruction-tuned large language model. For this purpose, we integrate our G-Star sampler into the **Dream-Instruct 7B** (Ye et al., 2025b) model and evaluate it on a diverse suite of complex downstream benchmarks. We establish our baseline by evaluating the Dream-Instruct model with the authors' official configuration. As shown in Table 1, our reproduced scores vary slightly from the originally published results and serve as the direct point of comparison for our method.

For our approach, we augment the Dream-Instruct baseline

*Table 1.* Downstream benchmark results for Dream-Instruct 7B. The best result is marked in **bold**.

|  | Dream-Ins. (Paper) | Dream-Ins. (Reproduced) | + G-Star (Ours) |
|---|---|---|---|
| MMLU | 67.0 | 69.9 | **71.2** |
| MMLU-PRO | 43.3 | 46.9 | **47.9** |
| GSM8K | 81.0 | 81.5 | **81.6** |
| GPQA | 33.0 | 31.0 | **32.8** |
| HumanEval | 55.5 | 53.7 | **54.9** |
| MBPP | 58.8 | 58.0 | **59.4** |
| IFEval | 62.5 | 56.4 | **59.3** |

*Table 2.* Conditional code generation results on the Conala benchmark for different samplers and step counts. All methods use the same conditional MDLM generator; Qwen2.5B-Coder is used only as a frozen external evaluator to compute conditional perplexity. **Best** and second-best results are highlighted.

| Algorithm | Qwen2.5B-Coder ppl $\downarrow$ | | |
|---|---|---|---|
|  | 32 steps | 64 steps | 128 steps |
| MDLM | 29.8 | 25.5 | 26.7 |
| ReMDM-loop$_{\eta=0.02}$ | 30.1 | 25.0 | 20.4 |
| ReMDM-cap$_{\eta=0.04}$ | 27.3 | 22.5 | 19.1 |
| G-Star-loop | 22.5 | **17.8** | 17.8 |
| G-Star+$_{t_{on}=0.3}$ | **20.4** | 18.9 | **16.4** |

with our G-Star sampler, integrating it via a loop-based refinement strategy. We keep the total number of diffusion steps identical to the baseline but designate 10% of them as refinement steps executed by G-Star at a specific noise level $\alpha_{on}$. The error predictor is configured for maximum parameter efficiency: we freeze the 7B model's backbone and train only a lightweight classification head on the Tulu 3 (Lambert et al., 2024) dataset. Detailed configurations for each benchmark are provided in Appendix E.

As summarized in Table 1, our G-Star sampler yields consistent performance gains across all seven evaluated benchmarks. We observe noteworthy improvements on complex reasoning tasks such as MMLU (+1.3 points) and GPQA (+1.8 points), as well as on instruction following (IFEval, +2.9 points). It validates that highly capable models still benefit from a dedicated mechanism for targeted error correction, further enhancing their reasoning and generation capabilities.

### 5.2. Code Generation

We evaluate our method on conditional code generation using the Conala benchmark (Yin et al., 2018), where the task is to generate a Python snippet from a natural language prompt. We first train a conditional MDLM baseline on the ConaLa train split and then benchmark G-Star against the strongest off-the-shelf remasking strategy (ReMDM) identified in Section 4.5. Implementation details are provided in Appendix E.

Performance is measured by conditional perplexity under a frozen pre-trained Qwen2.5B-Coder model (Hui et al., 2024). Qwen2.5B-Coder is used strictly as an external evaluator: it does not generate code, train the conditional MDLM, train the error predictor, or provide guidance during sampling. Thus, Table 2 isolates the effect of the sampling strategy while keeping the generator fixed. This metric evaluates the fluency and semantic relevance of the generated code snippet with respect to the input prompt. As shown in Table 2, our G-Star sampler outperforms both the MDLM and ReMDM baselines, achieving a lower (better) condi-

tional perplexity. This confirms the effectiveness of our guided approach for structured generation tasks.

## 6. Conclusion

We introduced G-Star, a sampling method that enables efficient error correction for masked diffusion models. By using a trained error predictor to target tokens for revision, our method outperforms standard and stochastic refinement baselines like MDLM and ReMDM in computationally constrained, few-step generation regimes. We demonstrated its effectiveness and versatility across a wide range of tasks and validated its ability to enhance a state-of-the-art 7B instruction-tuned language model. The core contribution of our work is to show that targeted, intelligent refinement is a more principled and sample-efficient approach than unguided correction, paving the way for more practical and powerful discrete diffusion models.

## Acknowledgements

This research was supported in part through computational resources of HPC facilities at HSE University. The work was supported by the grant for research centers in the field of AI provided by the Ministry of Economic Development of the Russian Federation in accordance with the agreement 000000C313925P4E0002 and the agreement with HSE University № 139-15-2025-009.

## Impact Statement

This paper focuses on methodological advancements in machine learning. We do not foresee specific societal consequences beyond the general risks associated with the field that would necessitate immediate highlighting.

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

## Reproducibility Statement

To ensure the reproducibility of our work, we release the source code for our samplers and error predictor training at https://github.com/EgorShibaev/G-Star. All experimental details, including dataset preprocessing, model architectures, and specific hyperparameter configurations for every table and figure, are thoroughly documented in Appendix E. Furthermore, our experiments are built upon publicly available datasets (e.g., OpenWebText, Conala) and pre-trained model checkpoints to ensure our experimental setups are accessible and verifiable by the community.

## A. Theoretical Results

### A.1. Proof of Claim 1

In this section, we prove Claim 1: *The Variational Lower Bound (VLB) for the star-shaped process simplifies to a weighted cross-entropy objective, which has the same functional form as the standard masked diffusion objective but with different timestep-dependent weights.*

We begin with the standard VLB formulation for a non-Markovian process, which seeks to maximize the log-likelihood $\log p_\theta(\mathbf{x}_0)$:

$$\log p_\theta(\mathbf{x}_0) \geq \mathbb{E}_{q(\mathbf{x}_{1:T}|\mathbf{x}_0)}\left[\log\frac{p_\theta(\mathbf{x}_{0:T})}{q(\mathbf{x}_{1:T}\mid\mathbf{x}_0)}\right] =: \mathcal{L}_{\text{VLB}} \tag{7}$$

By parameterizing the generative process as $p_\theta(\mathbf{x}_{0:T}) = p(\mathbf{x}_T)\prod_{t=1}^{T} p_\theta(\mathbf{x}_{t-1} \mid \mathbf{x}_t)$ and defining our star-shaped forward process as $q(\mathbf{x}_{1:T} \mid \mathbf{x}_0) = \prod_{t=1}^{T} q(\mathbf{x}_t \mid \mathbf{x}_0)$, we can decompose the VLB:

$$\mathcal{L}_{\text{VLB}} = \mathbb{E}_q\left[\log p_\theta(\mathbf{x}_0 \mid \mathbf{x}_1) - \sum_{t=2}^{T}\text{KL}\left(q(\mathbf{x}_{t-1}\mid\mathbf{x}_0)\big\|p_\theta(\mathbf{x}_{t-1}\mid\mathbf{x}_t)\right)\right] - \text{KL}(q(\mathbf{x}_T\mid\mathbf{x}_0)\|p(\mathbf{x}_T)). \tag{8}$$

The final term is a constant with respect to the model parameters $\theta$ and can be ignored during optimization. The first term, $\mathbb{E}_q[\log p_\theta(\mathbf{x}_0 \mid \mathbf{x}_1)]$, is a reconstruction term, which already matches the form of a cross-entropy loss. Our goal is to show that the summation of KL divergence terms can also be simplified into this form.

Let's analyze a single KL divergence term from the summation for a given timestep $t$:

$$\mathcal{L}_t = \text{KL}\left(q(\mathbf{x}_{t-1}\mid\mathbf{x}_0)\big\|p_\theta(\mathbf{x}_{t-1}\mid\mathbf{x}_t)\right). \tag{9}$$

We substitute the definitions for the distributions involved:

- The true posterior is $q(\mathbf{x}_{t-1} \mid \mathbf{x}_0) = \text{Cat}(\mathbf{x}_{t-1}; \alpha_{t-1}\mathbf{x}_0 + (1-\alpha_{t-1})\mathbf{m})$.

- The model's reverse transition is $p_\theta(\mathbf{x}_{t-1} \mid \mathbf{x}_t) = q(\mathbf{x}_{t-1} \mid \mathbf{x}_0 = \hat{\mathbf{x}}_0)$, where $\hat{\mathbf{x}}_0 = f_\theta(\mathbf{x}_t, t)$. This gives $p_\theta(\mathbf{x}_{t-1} \mid \mathbf{x}_t) = \text{Cat}(\mathbf{x}_{t-1}; \alpha_{t-1}\hat{\mathbf{x}}_0 + (1-\alpha_{t-1})\mathbf{m})$.

The KL divergence is therefore between two categorical distributions of the same family, both of which are interpolations between a one-hot vector (the true $\mathbf{x}_0$ or the predicted $\hat{\mathbf{x}}_0$) and the mask token $\mathbf{m}$. The KL term becomes:

$$\mathcal{L}_t = \sum_{v=1}^{|V|}(\alpha_{t-1}x_{0,v} + (1-\alpha_{t-1})m_v)\log\frac{\alpha_{t-1}x_{0,v} + (1-\alpha_{t-1})m_v}{\alpha_{t-1}\hat{x}_{0,v} + (1-\alpha_{t-1})m_v} \tag{10}$$

Since $\mathbf{x}_0$ is a one-hot vector corresponding to a non-mask token (let's say at index $k$), and $\mathbf{m}$ is a one-hot vector for the mask token, this sum simplifies significantly.

$$\mathcal{L}_t = \alpha_{t-1}\log\frac{\alpha_{t-1}}{\alpha_{t-1}\hat{x}_{0,k}} + (1-\alpha_{t-1})\log\frac{1-\alpha_{t-1}}{1-\alpha_{t-1}} \tag{11}$$

$$= \alpha_{t-1}\log\frac{1}{\hat{x}_{0,k}} + 0 \tag{12}$$

$$= -\alpha_{t-1}\log\hat{x}_{0,k} \tag{13}$$

Since $\hat{x}_{0,k}$ is the probability assigned by the model $f_\theta(\mathbf{x}_t, t)$ to the true token, this is precisely the negative log-likelihood, or cross-entropy loss:

$$\mathcal{L}_t = -\alpha_{t-1} \log p_\theta(\mathbf{x}_0 \mid \mathbf{x}_t). \tag{14}$$

Now, we can substitute this simplified form back into the full VLB expression from Eq. equation 8. The loss to be minimized (the negative VLB) is:

$$\mathcal{L}_{\text{final}} = -\mathcal{L}_{\text{VLB}} \approx \mathbb{E}_q \left[ -\log p_\theta(\mathbf{x}_0 \mid \mathbf{x}_1) + \sum_{t=2}^{T} \mathcal{L}_t \right] \tag{15}$$

$$= \mathbb{E}_q \left[ -\log p_\theta(\mathbf{x}_0 \mid \mathbf{x}_1) - \sum_{t=2}^{T} \alpha_{t-1} \log p_\theta(\mathbf{x}_0 \mid \mathbf{x}_t) \right] \tag{16}$$

This is a sum of weighted cross-entropy losses. By writing this as a single expectation over a timestep $t$ sampled uniformly from $\{1, \ldots, T\}$, we get:

$$\mathcal{L}_{\text{final}} = \mathbb{E}_{t \sim \mathcal{U}, \mathbf{x}_0, \mathbf{x}_t} \left[ -w'_t \log p_\theta(\mathbf{x}_0 \mid \mathbf{x}_t) \right] \tag{17}$$

where $w'_t$ are new, time-dependent weights derived from the coefficients (e.g., $w'_1 = 1$, and $w'_t = \alpha_{t-1}$ for $t > 1$, before normalization). This confirms that the training objective for the star-shaped process simplifies to the same functional form as the standard masked diffusion objective, completing the proof.

### A.2. Connection to ReMDM (Wang et al., 2025)

It is instructive to analyze the relationship between our proposed method and ReMDM (Wang et al., 2025), a recent framework that enables error correction. To achieve this, ReMDM modifies the standard MDLM posterior Equation (2) by introducing a hyperparameter $\sigma_t$ that explicitly governs the probability of reverting already generated tokens back to the [MASK] state:

$$q(\mathbf{x}_{t-1} \mid \mathbf{x}_t, \mathbf{x}_0) = \begin{cases} \text{Cat}(\mathbf{x}_{t-1}; (1-\sigma_t)\mathbf{x}_0 + \sigma_t \mathbf{m}), & \text{if } \mathbf{x}_t \neq \mathbf{m} \\ \text{Cat}\left(\mathbf{x}_{t-1}; \frac{\alpha_{t-1} - (1-\sigma_t)\alpha_t}{1-\alpha_t}\mathbf{x}_0 + \frac{1-\alpha_{t-1} - (1-\sigma_t)\alpha_t}{1-\alpha_t}\mathbf{m}\right), & \text{if } \mathbf{x}_t = \mathbf{m} \end{cases} \tag{18}$$

Here, $\sigma_t \in [0, \min\{1, \frac{1-\alpha_{t-1}}{\alpha_t}\}]$ is a hyperparameter controlling the probability that an already unmasked token is re-masked. A key practical challenge of ReMDM is that $\sigma_t$ is governed by complex schedules requiring careful tuning (see Appendix C.1).

Notably, our star-shaped sampler is not merely an alternative heuristic but establishes a direct theoretical connection to this framework. Specifically, we show that our method is mathematically equivalent to ReMDM under a specific remasking schedule.

**Proposition A.1.** *The Star-Shaped Masked Diffusion sampler (Eq. 5) is equivalent to the ReMDM sampler when the remasking parameter is set to $\sigma_t = 1 - \alpha_{t-1}$.*

*Proof.* Substituting $\sigma_t = 1 - \alpha_{t-1}$ into Eq. 18 simplifies both cases of the conditional probability:

*Case 1 ($\mathbf{x}_t \neq \mathbf{m}$):* The probability of a token remaining unmasked is $1 - \sigma_t = 1 - (1 - \alpha_{t-1}) = \alpha_{t-1}$.

*Case 2 ($\mathbf{x}_t = \mathbf{m}$):* The probability of unmasking a token becomes:

$$\frac{\alpha_{t-1} - (1-\sigma_t)\alpha_t}{1-\alpha_t} = \frac{\alpha_{t-1} - \alpha_{t-1}\alpha_t}{1-\alpha_t} = \frac{\alpha_{t-1}(1-\alpha_t)}{1-\alpha_t} = \alpha_{t-1}. \tag{19}$$

In both cases, the probability of $\mathbf{x}_{t-1}$ being clean reduces strictly to $\alpha_{t-1}$, rendering the transition independent of the current state $\mathbf{x}_t$. This recovers the marginal distribution $q(\mathbf{x}_{t-1} \mid \mathbf{x}_0)$, which is precisely the definition of our star-shaped transition. $\square$

While ReMDM relies on heuristic schedules to balance stability and correction, our formulation embraces maximum flexibility (independence from $\mathbf{x}_t$), which we subsequently stabilize via the learned guidance $g_\phi$ rather than manual hyperparameter tuning.

# B. Related Works

## B.1. Discrete Diffusion

Discrete diffusion extends denoising ideas from continuous domains (Sohl-Dickstein et al., 2015; Ho et al., 2020; Song et al., 2020) to categorical spaces by defining token-level noising/posterior transitions. Early work formalized absorbing/structured corruption for tokens (Austin et al., 2021a; Campbell et al., 2022), while ratio-estimation and reparameterized objectives improved likelihoods and stability for text/code (Lou et al., 2024; Zheng et al., 2023). Alternative transport in discrete spaces is discrete flow matching (Gat et al., 2024; Campbell et al., 2024; Nisonoff et al., 2024). Practical samplers and correctors further enhance decoding (Lezama et al., 2023; Zhao et al., 2024), and analyses clarify properties of absorbing processes and conditional distributions (Ou et al., 2024).

## B.2. Text Latent Diffusion

A complementary line runs diffusion in continuous text spaces. Diffusion-LM denoises word embeddings and enables controllable text via plug-and-play guidance (Li et al., 2022). Two-stage latent approaches compress sequences with a pretrained autoencoder and diffuse in the compact latent space, improving quality and reducing the step budget across unconditional and conditional tasks (Lovelace et al., 2024; Meshchaninov et al., 2025; Shabalin et al., 2025).

## B.3. Masked Diffusion

Masked-token diffusion adopts BERT-style prediction with iterative unmasking for fast, parallel generation. MaskGIT established the paradigm on tokenized images with few refinement rounds (Chang et al., 2022). For language, simple absorbing-mask diffusion with a clean training recipe narrows the perplexity gap to autoregressive LMs and supports flexible (semi-)autoregressive decoding (Sahoo et al., 2024). A simplified continuous-time view yields a weighted cross-entropy ELBO and state-dependent masking schedules that improve text and discrete-image modeling (Shi et al., 2024). Inference-time revision/guidance further boosts quality: remasking enables iterative correction (Wang et al., 2025), discrete guidance improves controllability (Schiff et al., 2024; Nisonoff et al., 2024), informed correctors sharpen updates (Zhao et al., 2024), hybrids expand self-correction regimes (von Rütte et al., 2025), and analyses examine time-agnostic behavior and sampling (Zheng et al., 2024).

## B.4. Remasking Diffusion Model

To improve generation quality, several methods introduce mechanisms for revising earlier token decisions. The most common approach involves random re-masking, where a portion of the sequence is arbitrarily reset and regenerated (Campbell et al., 2022; Wang et al., 2025). While simple, this strategy suffers from low sample efficiency. Since the re-masking is non-selective, the model wastes computational resources regenerating correct tokens, often requiring significantly extended inference schedules to fix isolated mistakes.

Alternatively, some works propose modifying the diffusion framework itself to support self-correction (von Rütte et al., 2025; Havasi et al., 2025; Song et al., 2025). These methods adjust the training objective to handle iterative edits, but they come with major drawbacks. Primarily, they necessitate the expensive retraining of heavy foundation models. Furthermore, these models are typically trained on synthetic noise distributions, such as uniform token replacement. This creates a discrepancy: detecting random vocabulary swaps is a much simpler task than correcting the sophisticated, structure-preserving errors that diffusion models actually produce.

A third direction explores confidence-based filtering, which seeks to identify errors using the model's own likelihood scores without additional training (Peng et al., 2025; Zheng et al., 2023). However, standard masked diffusion models are optimized for reconstruction, not calibration. As a result, their confidence scores on unmasked tokens are often poorly correlated with actual generation quality, making them an unreliable signal for targeted refinement.

## B.5. Large Language Diffusion Models

Recent efforts include blockwise decoding (Arriola et al., 2025), inference-time scaling (Ma et al., 2025; Singhal et al., 2025), and domain-focused large models for reasoning/coding (Nie et al., 2025; Zhao et al., 2025). Prominent systems include *LLaDA* (Nie et al., 2025), *Dream 7B* (Ye et al., 2025a), and *DiffuCoder* (Gong et al., 2025).

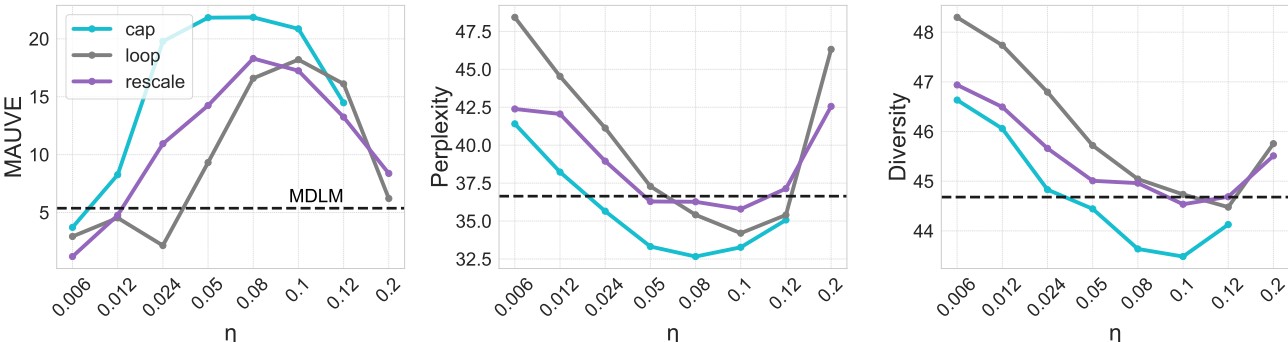

*Figure 6.* **Performance of ReMDM as a function of the hyperparameter** $\eta$**.** Results are shown for three different remasking schedules. The dashed line indicates the performance of the baseline MDLM sampler. The plots reveal a high sensitivity to the choice of $\eta$, with suboptimal values often performing worse than the baseline.

## C. Additional Results

### C.1. Hyperparameter Tuning in ReMDM

A significant practical limitation of the ReMDM sampler is its high sensitivity to the remasking hyperparameter, $\eta$. This sensitivity makes the sampler's performance brittle and necessitates a costly, non-trivial tuning process to achieve any benefit over simpler baselines. In this section, we empirically quantify this dependency and demonstrate that the optimal configuration for $\eta$ is not universal, but must be determined independently for each remasking schedule.

To analyze this, we conduct an ablation study on the value of $\eta$ for three different remasking schedules proposed by the authors: 'cap', 'loop', and 'rescale'. The experiments are performed on the OWT dataset, generating sequences of length 512 with 128 sampling steps.

The results, presented in Figure 6, confirm our hypothesis and reveal two significant drawbacks of the ReMDM approach. First, performance across all metrics (MAUVE, Perplexity, and Diversity) is extremely sensitive to the choice of $\eta$. As the plots demonstrate, the relationship is non-monotonic and exhibits a "sweet spot"; a small deviation from this optimal value can cause a dramatic drop in performance. Crucially, an improper configuration of $\eta$ can render the remasking mechanism actively detrimental, with performance falling significantly below that of the standard, non-refining MDLM baseline.

Second, the optimal value for $\eta$ is not universal but must be independently and carefully tuned for each remasking schedule. Our ablation reveals that the optimal setting for the 'cap' and 'rescale' schedules is $\eta = 0.08$, whereas the 'loop' schedule achieves its peak performance at $\eta = 0.1$. This necessity for an extensive, per-schedule hyperparameter search represents a significant practical limitation, as it requires numerous runs to find a configuration that provides a tangible benefit over simpler baselines. This motivates our work on a star-shape sampler that is inherently more robust and efficient.

### C.2. Ablation on Loop Size

As previously described, refinement process consists of: (1) an initial drafting phase with MDLM, (2) an iterative refinement "loop" at a fixed noise level, and (3) a final completion phase with MDLM. In this section, we investigate how the number of steps allocated to the refinement loop (the "loop size") affects generation quality.

**Setup.** For this experiment, we generate 512-token sequences from OWT. The steps are allocated as follows: 115 steps for the initial MDLM draft, 13 steps for the final completion, and a variable number of steps for the central refinement loop. We vary them across a predefined grid and compare the performance of our guided sampler (G-Star-loop) against its unguided counterpart (Star-loop).

**Results.** The results, presented in Figure 7, reveal several key dynamics of the refinement process.

First, for both samplers, increasing the number of refinement steps generally leads to higher-quality text, as evidenced by a monotonic decrease in perplexity and an initial rise in the MAUVE score. This confirms the efficacy of iterative refinement. However, our guided G-Star-loop is substantially more sample-efficient, achieving a much steeper improvement curve. It

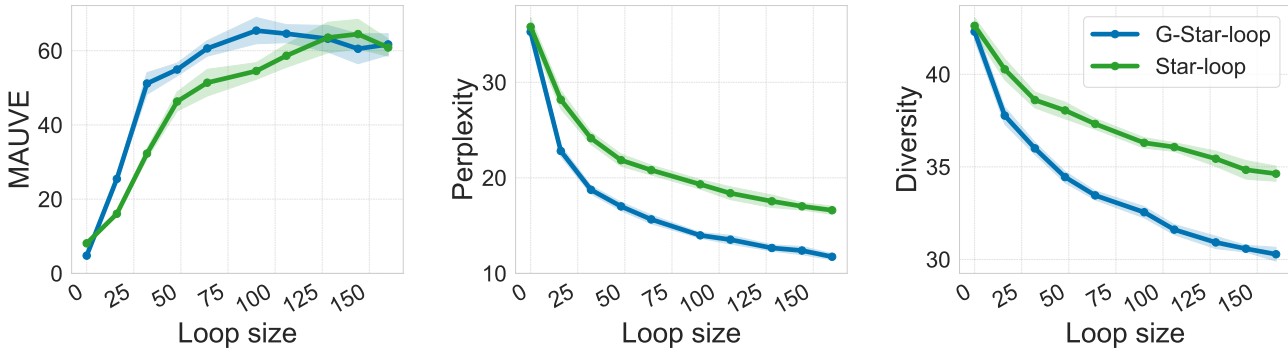

*Figure 7.* Performance as a function of the refinement loop size. Increasing the refinement budget generally improves quality (lower PPL, higher MAUVE) but reduces diversity. Our guided G-Star-loop demonstrates a much steeper rate of improvement, achieving higher quality with fewer steps. The MAUVE score eventually peaks and declines as the loss of diversity outweighs quality gains.

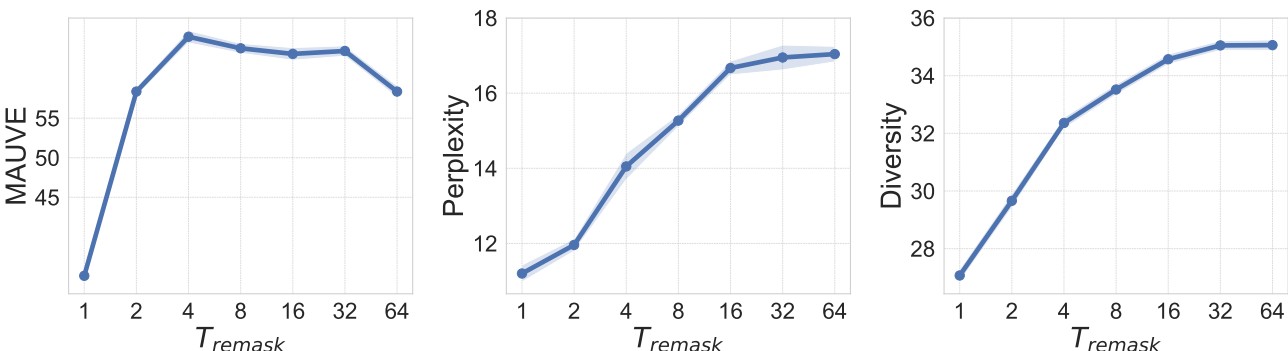

*Figure 8.* **Performance as a function of the error predictor temperature** ($T_{\text{remask}}$). The plots reveal a clear trade-off: lower temperatures improve quality (PPL) at the cost of diversity, while higher temperatures increase diversity at the cost of quality. The MAUVE score, which balances both, peaks at an optimal temperature of $T \approx 4 - 32$.

consistently reaches a higher quality ceiling with fewer refinement steps compared to the unguided Star-loop.

Second, the refinement process introduces a clear trade-off between quality and diversity. As shown in the rightmost panel, a larger loop size consistently leads to a reduction in sample diversity for both methods. This can be interpreted as the model converging towards higher-quality modes in the data distribution, pruning away "noisy" or less coherent generations, but at the risk of reducing overall variety.

Finally, this quality-diversity tension directly explains the behavior of the MAUVE score. As MAUVE balances both aspects, it initially rises with perplexity improvements but eventually peaks and begins to decline as the loss in diversity becomes too significant. This phenomenon is not an artifact of a specific sampler but appears to be an inherent property of intensive iterative refinement itself: with enough steps, any refinement process will inevitably improve perplexity at the cost of diversity.

### C.3. The Quality-Diversity Trade-Off: Tuning the Error Predictor Temperature

We analyze the effect of the error predictor's temperature, $T_{\text{remask}}$, a hyperparameter that scales the logits from $g_\phi$ before sampling. This temperature effectively controls the stochasticity of the remasking process, acting as an intuitive control knob for the sampler's behavior. The experiment is conducted on the OWT dataset (512 tokens), using our guided sampler with the parameter-efficient error predictor configuration (a frozen 12-block MDLM backbone with a trainable classification head).

The results, presented in Figure 8, reveal a clear and non-monotonic relationship between the predictor temperature and the overall generation quality as measured by MAUVE. This behavior is a direct consequence of an underlying trade-off between sample quality (Perplexity) and Diversity, which the temperature directly controls.

*Table 3.* Results for an error predictor trained on OWT and evaluated on out-of-domain datasets. For each evaluation dataset, we report the validation perplexity (PPL val) of the OWT-pretrained diffusion model and the binary classification metrics of the error predictor.

| Evaluation dataset | Domain | PPL val | Accuracy | AUC-ROC |
|---|---|---|---|---|
| OWT | General | $\leqslant 22.89$ | 0.88 | 0.94 |
| TinyStories | Stories | $\leqslant 12.72$ | 0.88 | 0.98 |
| OpenWebMath | Math | $\leqslant 33.72$ | 0.85 | 0.92 |
| CNN/DailyMail | News | $\leqslant 25.69$ | 0.88 | 0.94 |
| The Stack | Python code | $\leqslant 31.96$ | 0.84 | 0.90 |

At low temperatures (e.g., $T = 1$), the predictor's output becomes more deterministic, focusing the remasking on a small set of tokens with the highest predicted error probability. This leads to highly precise corrections of the most obvious errors, resulting in the best perplexity scores (middle panel). However, this precision comes at the cost of significantly reduced diversity (right panel), as the sampler explores a much narrower set of possible revisions.

Conversely, as the temperature increases, the error probabilities become more uniform. In this regime, the guided sampler's behavior converges towards that of the unguided, Star-loop sampler. This predictably increases sample diversity but degrades perplexity, as the error correction is no longer targeted and becomes less effective.

The MAUVE score, which balances both quality and diversity, peaks at a temperature of $T \approx 4 - 32$. At this point, the sampler achieves an optimal balance between precise error correction and sufficient generative variety. This analysis highlights that the predictor temperature serves as an important lever for controlling the generation process, allowing practitioners to tailor the sampler's behavior: lower temperatures can be used for high-fidelity tasks where correctness is paramount, while higher temperatures may be preferable for creative tasks where diversity is the primary goal.

## C.4. Robustness of the Error Predictor

In the experiments above, the error predictor was both trained and evaluated on the same dataset. An important question, however, is how sensitive the predictor is to the choice of training data and to what extent it generalizes to unseen domains. To investigate this, we trained the error predictor on the OWT dataset and evaluated it on several datasets spanning diverse domains:

- TinyStories (Eldan & Li, 2023): a synthetic dataset of short stories written in simple English at the comprehension level of a typical 3–4-year-old child.

- OpenWebMath (Paster et al., 2023): a dataset of high-quality mathematical text filtered from Common Crawl.

- CNN/DailyMail (Hermann et al., 2015): a dataset of news articles authored by professional journalists at CNN and the Daily Mail.

- The Stack (Kocetkov et al., 2022): a large-scale code dataset containing over 6 TB of source code in 358 programming languages, from which we use only the Python subset.

To assess both the diffusion model's performance and the error predictor's generalization, we randomly sampled 10,000 examples from each evaluation dataset. We analyze the diffusion model's behavior on these new domains by reporting its validation perplexity (PPL val). We then evaluate the OWT-trained error predictor's ability to identify these errors using two standard binary classification metrics: Accuracy (at a 0.5 probability threshold) and AUC-ROC. The AUC-ROC score is particularly relevant as it measures the predictor's quality across all thresholds, which is crucial given that our G-Star sampler does not rely on a fixed threshold.

The results, presented in Table 3, show several key trends. The TinyStories dataset appears to be the simplest case for both the diffusion model and the error predictor, which is evident from its minimal validation perplexity (12.72) and the predictor's near-perfect AUC-ROC score (0.98). Conversely, the CNN/DailyMail dataset seems closest to the OWT training data, as its validation perplexity (25.69) is near the OWT baseline (22.89), and it achieves an identical AUC-ROC score (0.94). Unsurprisingly, the most challenging domains for both models are the specialized OpenWebMath and The Stack

(Python code) datasets, which show higher perplexity and slightly lower predictor performance. Overall, however, the drop in predictor quality across these diverse domains is not severe. This suggests that the error predictor successfully learns general patterns of diffusion errors, allowing it to generalize effectively even when trained on only a single dataset.

## D. Computational Overhead

**Time Overhead.** We quantify the deployment cost of the different samplers by measuring end-to-end wall-clock latency on OpenWebText with sequence length $L = 512$ and $T \in \{128, 256, 512\}$ diffusion steps. All runs use batch size 1 on a single H200 GPU. Table 4 reports, for each method, the total generation time per 512-token sample and the corresponding number of effective Transformer forward passes (NFEs), where one NFE denotes a single pass of the full backbone over a length-$L$ sequence.

As autoregressive (AR) baselines we use a GPT-2 small (Radford et al., 2019) whose parameter counts are matched to MDLM backbone. We consider a latency-optimized AR model with key–value (KV) caching, and a variant that recomputes the full prefix at every step ("AR-w/o KV"). The latter is closer to our diffusion setting, where each update operates on the full sequence, and helps disentangle the effect of cache reuse from the intrinsic cost of masked diffusion. In principle, similar KV-based accelerations (e.g., (Wu et al., 2025) could also be adapted for masked diffusion; see Appendix H for a broader discussion of diffusion speed-up techniques.

For the diffusion baselines, MDLM, ReMDM, P2 and Star all use the same backbone with no auxiliary networks. Their compute therefore coincides, with $\text{NFE}_{\text{MDLM}} = \text{NFE}_{\text{ReMDM}} = \text{NFE}_{\text{P2}} = \text{NFE}_{\text{Star}} = T$. In DDPD, both the denoiser and the refinement model are full Transformer backbones of the same depth and are executed at every diffusion step. As a result, each sampling step requires two full forward passes over the sequence, yielding an effective cost of $\text{NFE}_{\text{DDPD}} = 2T$.

While Star only changes the masking policy and introduces no extra passes, G-Star augments this baseline with guidance that is active only on a subset of the trajectory. Let $\Delta = t_{\text{off}} - t_{\text{on}}$ be the fraction of guided steps, and let $D$ denote the number of Transformer blocks in the backbone while $B$ is the number of blocks used by the predictor. Measured in units of a full-depth pass, each guided step then contributes $(B/D)$ additional NFEs, so the total cost is

$$\text{NFE}_{\text{G-Star}} = T + T\Delta\frac{B}{D} \;=\; \left(1 + \Delta\frac{B}{D}\right)T.$$

In our configurations, the backbone has $D = 12$ blocks. G-Star-loop$_{12\text{B}}$ uses a full-depth predictor ($B = 12$) with $t_{\text{on}} = 0.55, t_{\text{off}} = 0.05$, giving NFE $= 1.5T$; G-Star-loop$_{1\text{B}}$ uses a single-block predictor ($B = 1$) with the same $t_{\text{on}} = 0.55, t_{\text{off}} = 0.05$, yielding NFE $= \left(1 + 0.5 \cdot \frac{1}{12}\right)T \approx 1.04T$; and G-Star$^+$ uses a full-depth predictor ($B = 12$) with $t_{\text{on}} = 0.2, t_{\text{off}} = 0$, giving NFE $= 1.2T$. The measured wall-clock times in Table 4 closely follow these ratios: the G-Star variants incur a controlled $(1 + \Delta B/D)$-factor overhead.

**Memory Overhead.** Star uses exactly the same backbone and parameterization as the underlying MDLM and therefore has no memory overhead. For G-Star, peak activation memory is essentially unchanged: the predictor operates on the current logits and is run sequentially after the base diffusion step, so we do not need to keep additional large activations in memory, only per-token error scores and the index set of remasked positions.

The remaining overhead comes from parameters. In the G-Star-loop$_{12\text{B},F}$ variant we store an additional full 12-block Transformer as the predictor. In the G-Star-loop$_{1\text{B},F}$ variant we only add a single Transformer block. In the parameter-efficient G-Star-loop$_{12\text{B},H}$ variant we do not store a second backbone at all and only add a small token-wise classification head on top of the existing model.

## E. Implementation Details

### E.1. Unconditional Text Generation on OpenWebText

For this experiment, we closely follow the original MDLM setup for the backbone, and only add a linear head for error predictor training.

**Error Predictor Head Training.** The error predictor head, $g_\phi$, was trained for 50,000 steps on a single H200 GPU approximately 48 hours using a global batch size of 512. For optimization, we used AdamW with a learning rate of 1e-4. The learning rate was managed by a constant schedule with warmup with 2,500 warmup steps.

*Table 4.* Wall-clock generation time and number of Transformer forward passes (NFEs) per 512-token sample on OpenWebText (batch size 1). Times are reported as mean $\pm$ standard deviation in seconds.

| Method | Steps $T$ | Time [s] | NFEs |
|---|---|---|---|
| AR (KV cache) | 512 | $2.28 \pm 0.03$ | 512 |
| AR (no KV cache) | 512 | $2.51 \pm 0.01$ | 512 |
| MDLM / ReMDM / P2 / Star | 128 | $2.26 \pm 0.09$ | 128 |
| DDPD | 128 | $4.63 \pm 0.11$ | 256 |
| G-Star-loop$_{1B,F}$ | 128 | $2.60 \pm 0.10$ | 133 |
| G-Star-loop$_{12B,H}$ | 128 | $3.43 \pm 0.10$ | 192 |
| G-Star-loop$_{12B,F}$ | 128 | $3.43 \pm 0.10$ | 192 |
| G-Star+$_{t_{\mathrm{on}}=0.2,12B}$ | 128 | $2.71 \pm 0.10$ | 154 |
| MDLM / ReMDM / P2 / Star | 256 | $4.56 \pm 0.03$ | 256 |
| DDPD | 256 | $9.46 \pm 0.07$ | 512 |
| G-Star-loop$_{1B,F}$ | 256 | $5.24 \pm 0.02$ | 267 |
| G-Star-loop$_{12B,H}$ | 256 | $6.96 \pm 0.07$ | 384 |
| G-Star-loop$_{12B,F}$ | 256 | $6.96 \pm 0.07$ | 384 |
| G-Star+$_{t_{\mathrm{on}}=0.2,12B}$ | 256 | $5.49 \pm 0.06$ | 308 |
| MDLM / ReMDM / P2 / Star | 512 | $9.16 \pm 0.07$ | 512 |
| DDPD | 512 | $18.89 \pm 0.09$ | 1024 |
| G-Star-loop$_{1B,F}$ | 512 | $10.44 \pm 0.08$ | 533 |
| G-Star-loop$_{12B,H}$ | 512 | $13.93 \pm 0.08$ | 768 |
| G-Star-loop$_{12B,F}$ | 512 | $13.93 \pm 0.08$ | 768 |
| G-Star+$_{t_{\mathrm{on}}=0.2,12B}$ | 512 | $10.84 \pm 0.11$ | 615 |

### E.2. Code Generation on Conala

**Dataset and Preprocessing.** We use the Conala benchmark (Yin et al., 2018), which contains Python code snippets paired with natural language intents. We construct our dataset splits as follows: the train set consists of 2,000 curated samples plus 594,000 samples from the mined subset; the hold-out set for training the error predictor contains 380 curated samples; and the test set contains 500 samples. All prompts and code snippets were tokenized using the gpt-2 tokenizer, with sequences truncated or padded to a maximum length of 128 tokens.

**MDLM Baseline Training.** Our baseline is a conditional Masked Diffusion Language Model (MDLM), following the 12-layer Transformer architecture of Sahoo et al. (2024). We employed a two-stage training procedure. First, the model was pre-trained for 50,000 steps on the full train set (mined and curated combined) with a batch size of 1024. Subsequently, the model was fine-tuned for an additional 10,000 steps exclusively on the curated portion of the training data, using a smaller batch size of 512. For both stages, we used the AdamW optimizer with a learning rate of 3e-4.

**Error Predictor Training.** The error predictor, $g_\phi$, for our G-Star sampler was trained on the hold-out split. We employed a parameter-efficient setup: the predictor's backbone consists of the full 12-layer transformer from our trained conditional MDLM with its weights frozen. We then added a single linear classification head on top of the final layer's token representations, and trained only this head to predict token-level errors, conditioned on the same prompts. We used the AdamW optimizer with a learning rate of 3e-4 and a batch size of 380. The model was trained for 500 steps. The training takes 1 hour on a single H200 GPU.

### E.3. Large-Scale Experiments

This section details the experimental setup used for the large-scale evaluation on the Dream-Instruct 7B model, with results presented in Table 1.

**Models and Benchmarks.** We use the publicly available Dream-Instruct 7B model as our base model. The evaluation is conducted on a diverse suite of seven benchmarks: MMLU (Hendrycks et al., 2020), MMLU-PRO, GSM8K (Cobbe et al., 2021), GPQA (Rein et al., 2023), HumanEval (Chen et al., 2021), MBPP (Austin et al., 2021b), and IFEval. The sequence length for each benchmark is specified in Table 5.

*Table 5.* Benchmark-specific sequence lengths and the noise level $\alpha_{on}$ for the refinement loop.

| Benchmark | Sequence Length ($L$) | $\alpha_{on}$ |
|---|---|---|
| MMLU | 128 | 0.88 |
| MMLU-PRO | 128 | 0.88 |
| GSM8K | 256 | 0.95 |
| GPQA | 128 | 0.88 |
| HumanEval | 768 | 0.98 |
| MBPP | 1024 | 0.98 |
| IFEval | 1280 | 0.98 |

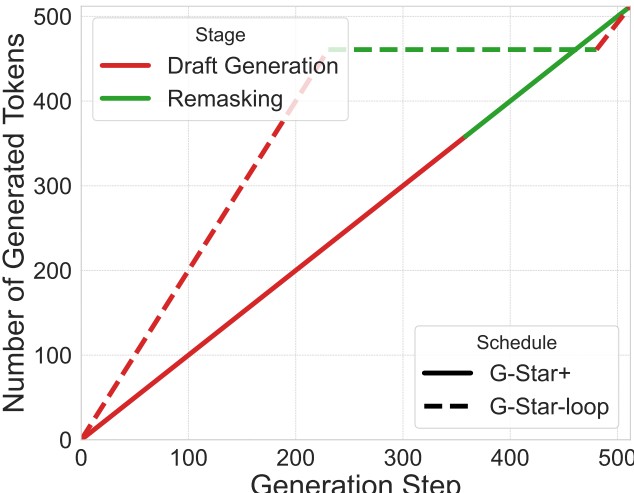

*Figure 9.* The plot illustrates the trajectory of the number of generated (unmasked) tokens across 512 generation steps. The G-Star+ schedule (solid line) follows a linear progression, dedicating the majority of steps to draft generation and applying remasking only towards the end of the trajectory. In contrast, the G-Star-loop schedule (dashed line) adopts an aggressive drafting strategy, rapidly producing a full candidate sequence (steep red slope) to maximize the computational budget allocated to the remasking phase (green plateau), enabling deep iterative refinement.

**Baseline Sampling Configuration.** Our reproduced baseline follows the official configuration from the Dream repository. The sampling process consists of a number of steps equal to the sequence length ($T = \text{seq\_len}$), where one mask is denoised at each step. The diffusion temperature is set to 0.1. The confidence score for selecting which token to unmask is calculated as the entropy of the predicted logits, as specified by their 'alg=entropy' setting.

**G-Star Sampler Configuration.** For our method, we augment the baseline setup by integrating a loop-based refinement strategy within the same computational budget. The total number of sampling steps is kept identical to the baseline ($T = \text{seq\_len}$), but we repurpose 10% of these steps for refinement. Specifically, 90% of the steps are used for standard progressive denoising, while the remaining 10% are dedicated to refinement loops where, at each step, we remask $N = 15$ tokens identified as errors by our predictor. All other parameters, such as the diffusion temperature (0.1) and the base confidence metric (entropy), are kept identical to the baseline for a fair comparison. The error predictor temperature was set to 0.

**Error Predictor Training.** The error predictor $g_\phi$ for our G-Star sampler was trained on the Tulu 3 dataset (Lambert et al., 2024). We employed a parameter-efficient strategy: the predictor's backbone consists of the full, frozen Dream-Instruct 7B model. We added a lightweight classification head, consisting of an RMSNorm layer and a linear layer, and trained only this head. The predictor was trained for 70k steps on 8 H200 GPUs over 24 hours. We used a global batch size of 128 and the AdamW optimizer with a learning rate of $3 \times 10^{-4}$, $\beta_1 = 0.9$, $\beta_2 = 0.999$, and a constant learning-rate schedule with a warmup of 5000 steps.

## F. Visualizing the Refinement Process

To provide a qualitative and intuitive understanding of the difference between our guided sampler and its unguided counterpart, we visualize their remasking behavior over the course of a full generation. The following figures illustrate the set of masked tokens (orange dots) at each step of the generation process for a 512-token sequence generated in 256 steps. Both processes are divided into two distinct phases: an initial MDLM drafting phase (steps 0-113), where tokens are progressively unmasked, a subsequent refinement phase (steps 114-240), where remasking occurs, and final MDLM generation phase (steps 241-256).

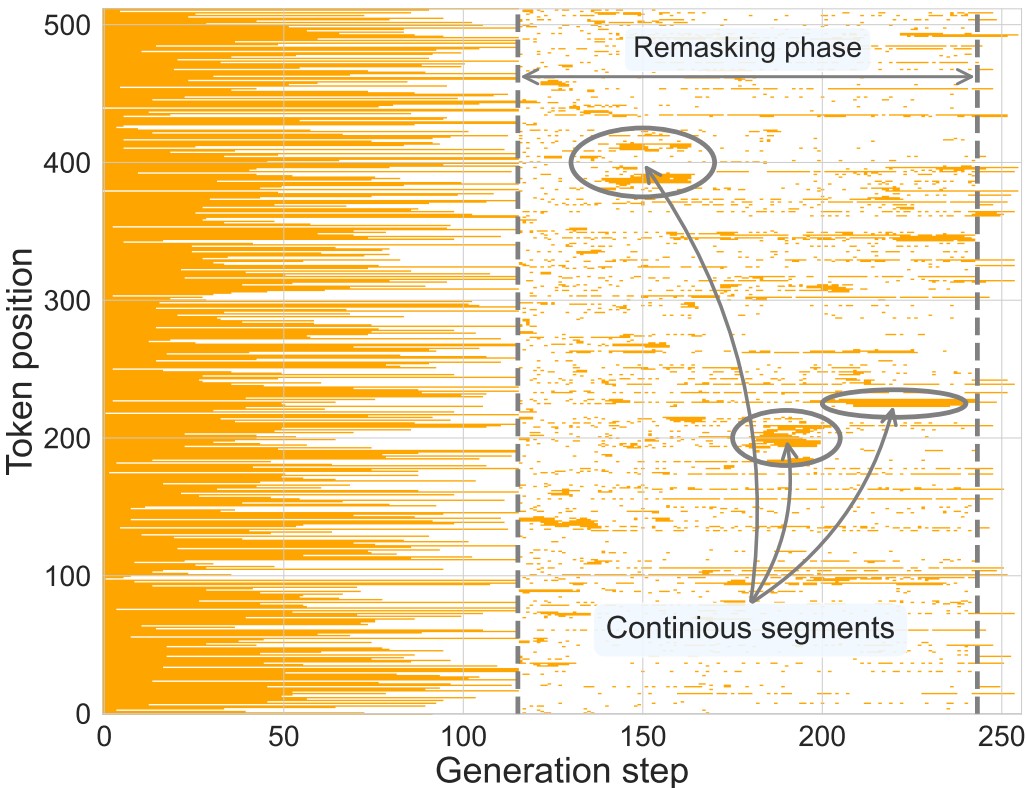

*Figure 10.* Remasking pattern of the guided G-Star-loop sampler. The plot visualizes the masked token positions (orange dots) at each generation step. In contrast to the unguided sampler, our guided approach exhibits a highly structured remasking pattern. The error predictor directs the sampler to focus on specific, clustered regions of the text. This often results in the selection of **contiguous segments** for revision, as highlighted in the figure. This ability to perform coordinated, phrase-level corrections is a key advantage of our targeted approach.

**Analysis of Remasking Strategies.** A direct comparison of Figure 11 and Figure 10 reveals the fundamental difference between the two refinement strategies. The unguided sampler operates via a stochastic, unstructured process, treating all tokens as equally likely candidates for revision. In stark contrast, our guided sampler demonstrates an intelligent and structured approach. The error predictor identifies and clusters likely errors, enabling the sampler to perform more meaningful revisions. The emergence of "continuous segments" in the guided plot is particularly significant; it provides strong qualitative evidence that our method moves beyond simple token-level fixes and is capable of performing coherent, phrase-level refinement, a feat that is extremely improbable under the indiscriminate selection of the unguided approach.

Figure 12 shows the generation trajectory of the DDPD sampler. Initializing from uniform noise introduces many errors, making the remasking process noisy. Nevertheless, several positions are remasked more than once.

Figure 13 shows the generation trajectory for the P2 self-planning sampler: after an initial period of remasking at the beginning of the remasking phase, no further remasking is performed, and the token configuration remains unchanged for the remainder of the generation process.

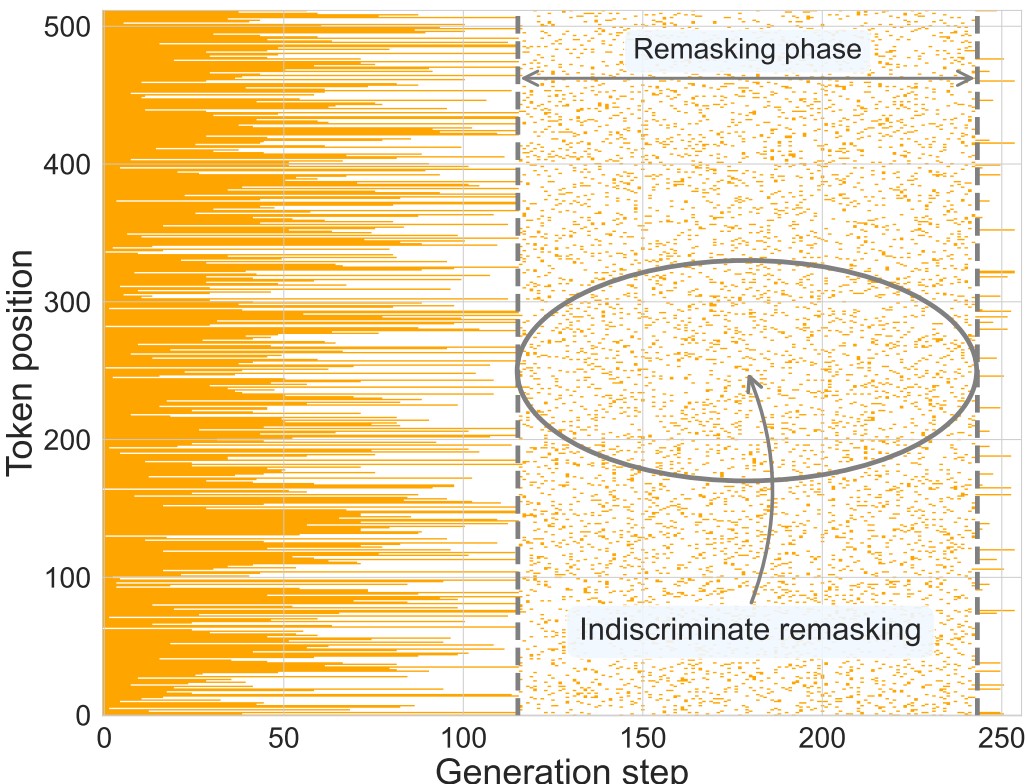

*Figure 11.* Remasking pattern of the Unguided Star-loop sampler. The plot visualizes the masked token positions (orange dots) at each generation step. During the remasking phase (steps 114-240), the pattern of selected tokens is scattered and visually resembles random noise. This illustrates the indiscriminate nature of the unguided approach, where every token position has a roughly equal probability of being revised at each step, without regard to the semantic or syntactic structure of the text.

**Text Generation Example of G-Star Sampler.** In addition, in this section, we provide examples of text generation using G-Star and the unguided Star. Figure 14 provides a visual snapshot of the refinement process, showing steps 90 through 95. Both G-Star+ (top) and the unguided Star+ (bottom) begin this phase with an identical text draft. The tokens highlighted in red are those selected for remasking at each step.

A clear difference in strategy is immediately visible. The unguided Star+ sampler (bottom) exhibits an unfocused, token-level remasking, selecting apparently random tokens for revision (e.g., steps 91 and 94). This indiscriminate approach is inefficient, as it may remask already correct tokens while failing to target problematic phrases. In stark contrast, G-Star+ (top) demonstrates a more structured and intelligent approach. Guided by the error predictor, it identifies and remasks semantically problematic regions. For example, in the transition from step 90 to 91, it targets the weak phrase "a period and mostly silent" for a coherent, phrase-level revision, resulting in "that we are". This targeted correction is highly unlikely to occur with the random sampling of Star+ and allows G-Star+ to perform more efficient, surgical edits to improve text quality.

## G. Practical Guidance for Hyperparameter Tuning

This section provides practical guidance for readers who wish to apply our method and select appropriate hyperparameters. The two most important parameters are the remasking schedule (i.e., when to apply G-Star) and the sampling temperatures.

**Remasking Schedule.** As we demonstrate in Section 4.2, text generation via masked diffusion can be broadly divided into two phases: an initial **context accumulation phase** and a subsequent **text refinement phase**. This two-stage structure aligns with practical observations from previous work on remasking, such as ReMDM (Wang et al., 2025). Based on this insight, we recommend enabling the remasking process (i.e., refinement) only toward the end of the generation, once the model has already formed a coherent draft. In our experiments, we typically activated the remasking schedule within the noise level range of $t \in [0.1, 0.3]$, for both the G-Star-loop and G-Star+ strategies. For optimal results on a specific task, we

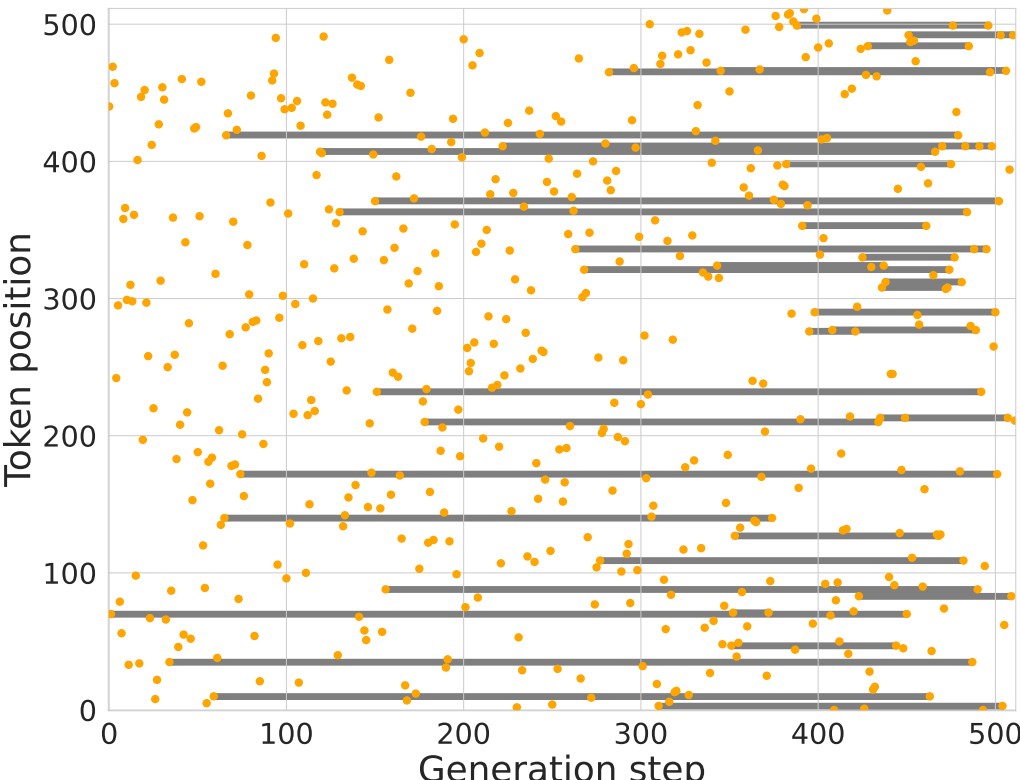

*Figure 12.* Remasking pattern of the DDPD sampler. The plot shows the token positions (orange dots) selected by the planner at each generation step. The selected-token pattern is largely scattered and visually resembles random noise, which is a consequence of initializing from a uniformly sampled sequence that introduces many errors. Despite this, some positions are still remasked more than once (indicated by gray segments).

recommend tuning this starting timestep $t$ as a key hyperparameter.

**Sampling Temperatures.** The second set of critical hyperparameters involves the diffusion and predictor temperatures. As discussed in Section 4.4, increasing the **diffusion model's temperature** produces more diverse and varied token predictions. This can be strategically advantageous: one can use a higher diffusion temperature to encourage *exploration* (proposing a wider set of token options), while relying on the error predictor to *filter* these proposals and retain only the correct ones. Separately, the **error predictor's temperature** (analyzed in Section C.3) controls its confidence. Lowering the predictor's temperature makes it more conservative, i.e., it will only remask tokens that it is highly confident are incorrect. However, since the predictor is not perfect and can also make mistakes, setting this temperature to an extremely low value (e.g., zero) may not be ideal. We recommend using a non-zero temperature to balance the predictor's precision and recall.

## H. Limitations

Despite its effectiveness, our method has three primary limitations. First, our framework is restricted to "in-place" token substitution and cannot perform insertion or deletion operations. This means that while the model can correct a token by changing its value (e.g., 'house' → 'home'), it cannot correct an error of omission by inserting a new token between two existing ones, as this would require shifting the entire subsequent sequence. Extending the framework to predict and apply "shift" or "insert/delete" operations is a promising direction for future work.

Second, the error predictor requires a separate, sequential training stage, which adds complexity to the overall training pipeline. This could potentially be addressed by exploring methods for jointly training the main diffusion model and the error predictor in an end-to-end fashion, which might also foster a tighter synergy between the generation and refinement processes.

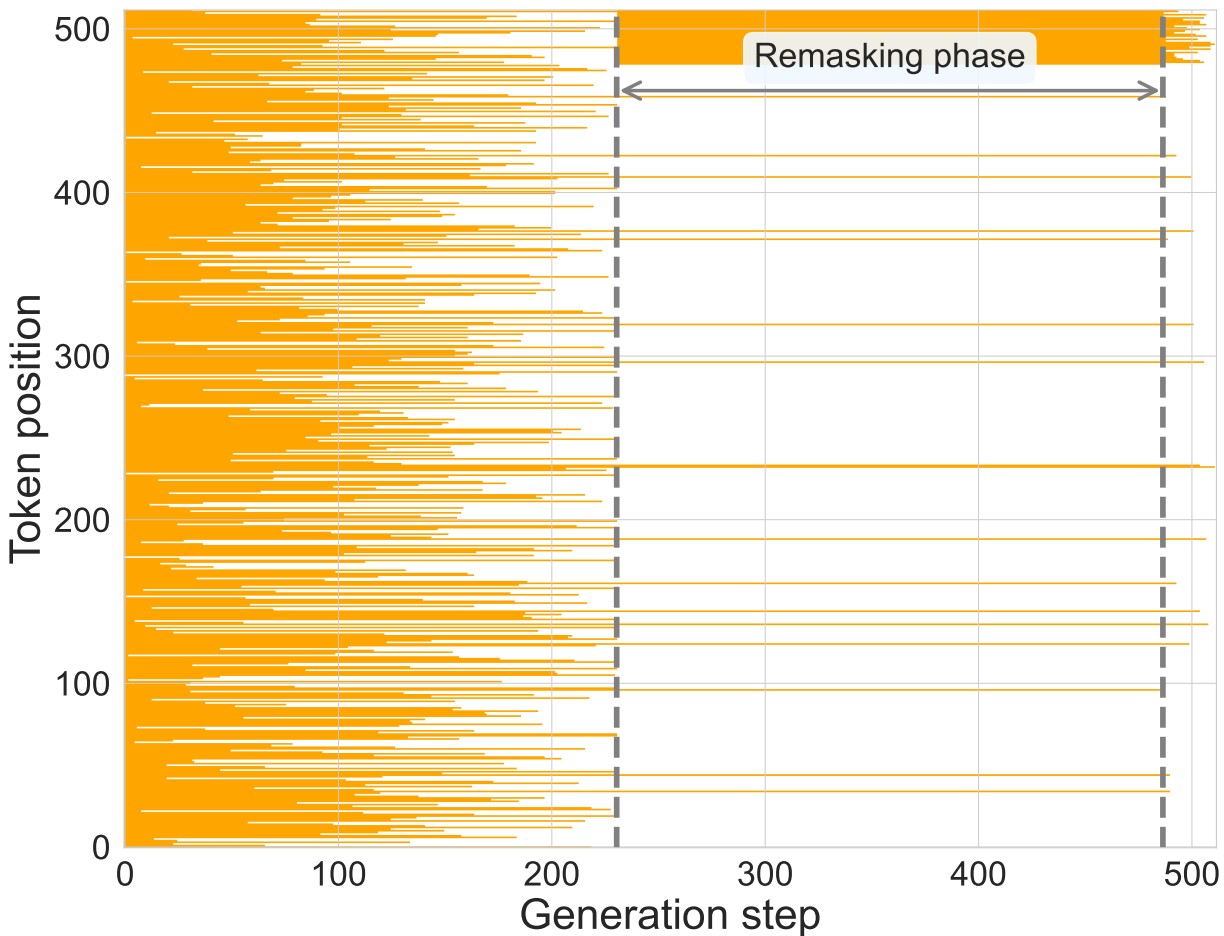

*Figure 13.* Remasking pattern of the P2 self-planning sampler. The plot visualizes the masked token positions (orange dots) at each generation step. During the remasking phase (steps 228-480), remasking is applied only in the initial steps; after this early period, no further remasking occurs and the token configuration remains fixed.

Third, the error predictor is trained on drafts obtained by denoising corrupted ground-truth sequences, but it is used at inference time on drafts produced along the model's own iterative sampling trajectory. This train–inference mismatch may affect the calibration of the predicted error scores. Our reduced-mismatch training variant slightly improved MAUVE while leaving perplexity and diversity nearly unchanged, but it roughly doubled the training cost because each update required additional denoising passes. Developing more scalable ways to expose the predictor to inference-like trajectories is an important direction for future work.

Finally, our study does not attempt to aggressively optimize inference throughput. All samplers are evaluated using a straightforward implementation that does not exploit recent acceleration techniques for diffusion LLMs, such as KV-caching (Wu et al., 2025), self-speculative decoding (Gao et al., 2025), or enhanced forms of parallel decoding like adaptive parallel decoding (Israel et al., 2025). Integrating G-Star with these complementary methods represents an exciting direction for future work and could further narrow the remaining gap to highly optimized autoregressive systems.

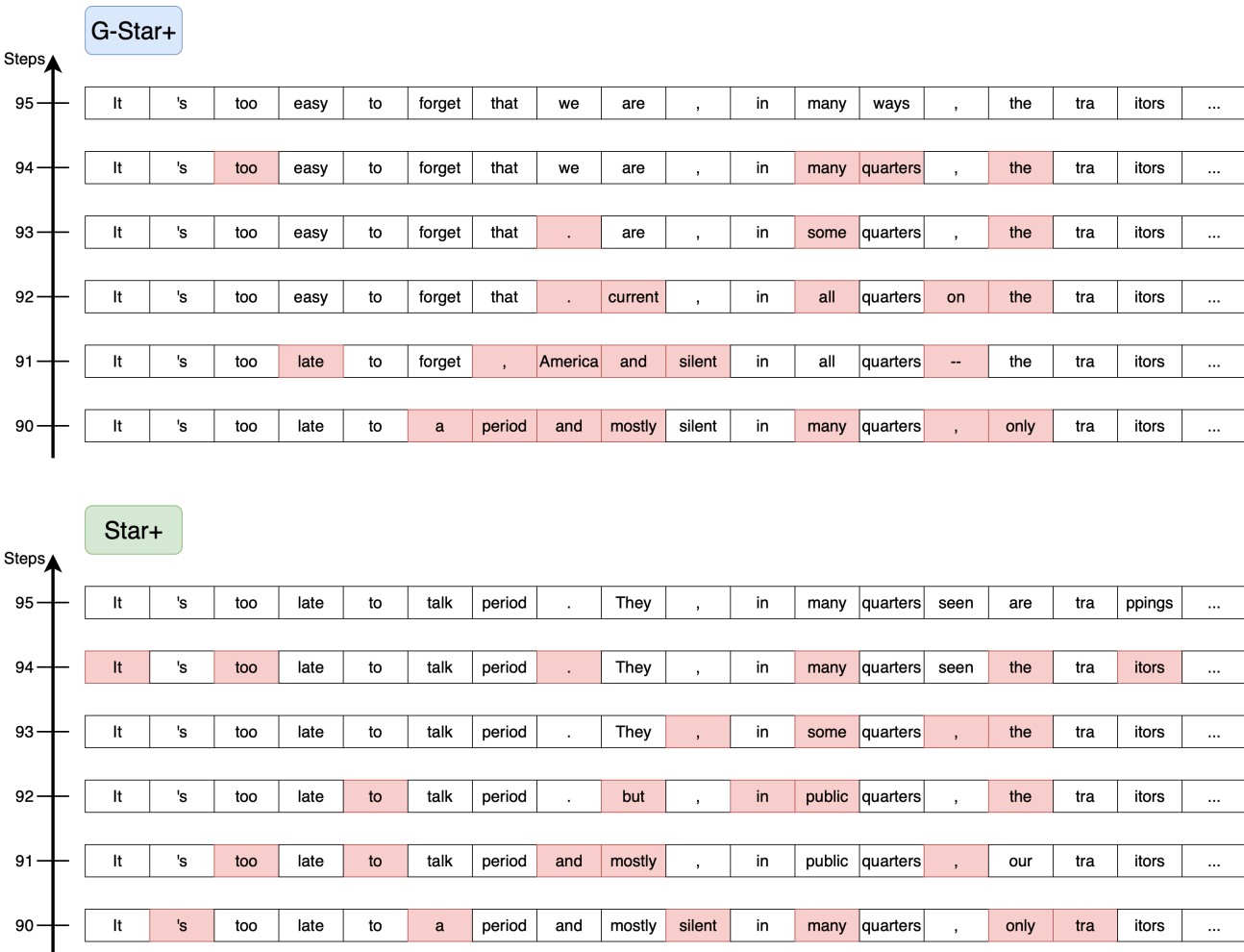

*Figure 14.* **Qualitative refinement trajectories.** We show steps 90–95 from a 128-step generation of a 512-token sequence. Both unguided Star+ (bottom) and guided G-Star+ (top) start this phase from the same text draft produced by standard MDLM sampling. The panels display the beginning of the sequence; tokens remasked at each step are highlighted in red. Starting from the same draft, the two samplers immediately diverge. Star+ applies scattered token-level remasking, including to tokens that already form plausible context. G-Star+ instead produces a more structured refinement trajectory, concentrating remasking on likely erroneous regions while preserving more of the coherent draft.

