# OpenReview forum: "Guided Star-Shaped Masked Diffusion"
_ICML.cc/2026/Conference — ICML 2026 regular_

### Official Review · Reviewer_3JKH · 2026-03-08

**Soundness:** 2
**Presentation:** 1
**Significance:** 2
**Originality:** 1
**Overall Recommendation:** 2
**Confidence:** 4

**Summary:**

This paper proposes methods to address limitations of current remasking strategies. A sampling algorithm followed by finetuning is used to improve sample quality and efficiency. Also, masks are updated in a learnable way to revise likely errors. The experiments are conducted on MDLM using a small number of sampling steps.

**Compliance With Llm Reviewing Policy:**

Affirmed.

**Ethical Review Concerns:**

Remasking can result in regressive performance besides efficiency gains, such as unsafe responses.

**Ethical Review Flag:**

Flag this paper for an ethics review.

**Final Justification:**

Considering the significance, novelty, and experimental quality, I would suggest rejecting this paper.
In the rebuttal, the authors have not provided new evidence to support the claim.

Limitation: Both writing and motivation are about a low-step regime. The authors state that they have conducted experiments on a 512-step setting. However, it is clear that the model performance is not convergent with 512 steps (Figure 3). Steps beyond 512 are the basic standard.

The proposed method is not training-free, which is a limitation compared with the majority of training-free baselines.

Novelty. The difference between the paper and the related works is marginal at the high level. It is just small changes. While the sampling is the core method, the paper also leverages a fine-tuning strategy. It is unclear which part results in the performance improvements.

Experiments: limited comparison with existing masking strategies and sampling strategies.

The paper starts with 3 limitations. It is unclear which one is the focus of the paper. The motivations have not been directly verified and the experiments also have not fullly resolve the limitation.

I also review the comments from other reviewers. To match the bar of ICML, I feel this paper needs a major revision to refine the claim, analyse the negative impact and cost, add more experiments, and improve writing.

**Key Questions For Authors:**

NA

**Limitations:**

The limitation of this work is that it only contribute the masked DM.

**Strengths And Weaknesses:**

Strengths

1. The limitations of current remasking strategies in masked DM are important.
2. The experiments indicate improvement in low-step regimes.

Weakness:

1. The first major limitation of this work is that it only addresses low-step generation. However, we usually care more about the ultimate performance in a high-step regime. It is well-known that a low number of steps only generates blurred images. There are many strategies proposed to improve efficiency. But none of the strategies only compares performance in a low-step regime.

2. Another major concern is the motivation. The paper introduces 3 limitations: computational inefficiency, unreliable confidence-based heuristics, and poor generalization. But none of them are empirically verified and reported. What is worse, none of the issues have been directly reported in the experiment.

3. The authors have mentioned two concurrent works. The reviewer is not sure which one is the first work, but the difference between these works is not significant enough. Besides these preprints, two published papers [1] hinder the novelty.

[1] Think while you generate: Discrete diffusion with planned denoising
[2] A reparameterized discrete diffusion model for text generation

4: The experiment quality is poor. While the sampling is the core method, the paper also leverages a fine-tuning strategy. It is unclear which part results in the performance improvements. Only MDLM is used. Datasets besides OWT should be introduced to strengthen the conclusion.

5 Again, it is unclear why the paper only reports results with low generation steps and low sampling steps. The paper needs to justify that the proposed method has no negative impact and substantial extra cost on the high-step regime.

6 The paper presentation is not professional, e.g., Figure 4 is a waste of space, and only 3 chars at the last line of the abstract.

---

> ### Author Rebuttal · Authors · 2026-03-27
>
> Thank you for your review. We respectfully believe that several of the main concerns do not accurately reflect what is already shown in the paper, particularly regarding the evaluated step regimes, the empirical support for our motivation, and the breadth of our experimental study. We address these points directly below.
>
> > **W1, W5.** On the claim that our work only addresses low-step generation.
>
> Our method is not limited to low-step generation, and the paper explicitly evaluates it beyond that regime. In Fig. 4a, we report G-Star-loop results for **128, 256, and 512 denoising steps**, while the baselines are evaluated with the standard **512-step** budget. This therefore includes the standard full-budget setting, not only reduced-step generation.
>
> Fig. 4a shows both points: **128-step** G-Star-loop already surpasses all **512-step** baselines on the quality-diversity frontier, and increasing our own budget to **256** and **512** steps further improves quality. It is therefore not accurate to describe the paper as addressing only low-step generation. The method is effective across step budgets, with the largest practical gains when the budget is limited.
>
> > **W2.** On the claim that the motivation is not empirically verified.
>
> The paper provides direct empirical support for all three motivations. For computational inefficiency, Fig. 3 shows that guided remasking consistently outperforms unguided Star+, demonstrating that stochastic remasking is markedly less efficient. For confidence-based heuristics, Sec. 2 explains why confidence on already clean tokens is not a reliable signal in masked diffusion, while Fig. 4a shows that our method outperforms confidence-based baselines such as P2-planner and RMD; Appendix F provides qualitative evidence of these differences in remasking behavior. For generalization, our predictor is trained on actual diffusion-model errors rather than synthetic random replacements, and Table 3 shows strong out-of-domain generalization across stories, news, math, and code.
>
> > **W3.** On novelty relative to prior work.
>
> Сontribution from prior confidence-based remasking and refinement-based methods is discussed in Sec. 2 / Appendix B.4, together with their limitations that motivate our method, and they are also included in our experimental comparison. The fact that our approach outperforms these baselines is part of the evidence for the value of our design.
>
> Our contribution is a practical formulation of **targeted refinement for pretrained masked diffusion models**. Prior remasking approaches improve revision from different angles, but they leave a common gap: stochastic remasking is inefficient, confidence-based remasking relies on uncalibrated signals on clean tokens, and learned refiners are often trained on synthetic corruption or require retraining the full backbone. Our method closes this gap by adapting the **star-shaped sampler** to masked diffusion in a form that works with **off-the-shelf MDLMs**, applying it specifically in the **late generation stage** where refinement is effective, and guiding it with a **lightweight predictor** trained on **real diffusion-model errors**. This combination is what distinguishes our approach from prior work.
>
> > **W4.** On the claim that the experimental evaluation is too narrow.
>
> We respectfully disagree with the statement that the paper only studies MDLM on OWT. OWT is used for controlled analysis of remasking strategies, but the paper also includes **large-scale experiments on Dream-Instruct 7B** (Table 1) and **code generation on Conala** (Table 2). Our comparisons are also not limited to MDLM alone: we compare against **MDLM, ReMDM, P2-planner, RMD, and DDPD**.
>
> > **W6.** On presentation quality.
>
> We will fix the abstract line-break artifact in the revision. We believe that Fig. 4 is important because it provides the appropriate basis for comparing methods across the full quality-diversity trade-off. Since the optimal denoiser temperature differs across methods, a single-point comparison can be misleading. The Pareto frontiers in Fig. 4 are therefore necessary for a fair comparison.
>
> > **W7.** On the statement that contributing only to masked diffusion may not be the right direction.
>
> We disagree that contributing to masked diffusion is itself a limitation. Masked diffusion is an important direction for parallel text generation, and one of its central weaknesses is the inability to revise earlier mistakes efficiently. Our paper addresses exactly this bottleneck and validates the approach beyond a toy setting, including large-scale language generation and code generation.
>
> ---
>
> Overall, we appreciate the reviewer’s time and feedback. We hope these clarifications resolve the main concerns, and we would be grateful if the reviewer would reconsider the assessment in light of the points above.

---

> > ### Author Rebuttal · Reviewer_3JKH · 2026-04-03
> >
> > Thanks for the response. However, there are inherent weaknesses that can not be addressed without a major revision and extensive experiments. My concerns regarding negative impacts and additional cost have not been directly addressed.
> >
> > W1&W5: The claim (the very first sentence) in your paper is a low-step regime. The model performance is not convergent at 512 steps.
> >
> > W2: The connection between the limitation and the experimental resutls are not directly related.
> >
> > W3: Novelty. The difference between the paper and the related works is marginal at the high level. It is just small changes.
> >
> > W4: Even though OWT is used for remasking strategies, only one dataset does not help the generalization of the conclusion.
> >
> > W6: presentation quality. Without a major revision, I can't see how the presentation can reach the bar of ICML.

---

> > > ### Author Response · Authors · 2026-04-07
> > >
> > > Thank you again for taking the time to read our rebuttal and for clarifying your remaining concerns.
> > >
> > > We understand that you may still remain unconvinced by the paper overall. However, we would like to respectfully note that several of the points in your acknowledgement still do not seem to reflect what is already evaluated in the submission.
> > >
> > > **W1 & W5.** We respectfully disagree that the paper only studies low-step generation. In Fig. 4a / Sec. 4.4, we evaluate G-Star-loop on **512-token OpenWebText generation** at **128, 256, and 512 denoising steps**, against baselines run with the standard **512-step** budget. This includes the standard full-budget regime, not only reduced-step generation. The same section also analyzes the quality-diversity trade-off across methods. In addition, Sec. 4.5 and Appendix D directly quantify the extra cost through latency and NFE measurements, so the additional cost is explicitly discussed in the paper.
> > >
> > > **W2.** We also respectfully disagree that the connection between the limitations and the experiments is not addressed. This connection is part of the design of Sec. 4: Fig. 3 isolates the efficiency gain of guided over unguided remasking, Fig. 4a compares against confidence-based and refinement-based baselines, Appendix F visualizes the different remasking behaviors, and Table 3 shows that the error predictor generalizes beyond the training domain. These experiments were intended precisely to test the motivations stated in the paper.
> > >
> > > **W3.** We understand that you may view the novelty as incremental. That said, we believe the distinction is more substantive than that characterization suggests. The related methods you pointed to are already discussed in the paper, and they are also included in our comparisons. Our contribution is a practical plug-and-play formulation of targeted refinement for **off-the-shelf masked diffusion models**, with a **lightweight predictor trained on real diffusion-model errors**. This combination is the main distinction of our method.
> > >
> > > **W4.** We respectfully disagree that the conclusions are based only on one dataset. OpenWebText is used for the controlled analysis of remasking strategies, but the paper also includes experiments beyond OWT in Sec. 5: large-scale evaluation on **Dream-Instruct 7B** and conditional **code generation on Conala**. So OWT is not the only empirical basis for the paper’s conclusions.
> > >
> > > **W6.** We accept that the presentation can still be improved. However, the review and acknowledgement provide very few concrete presentation issues beyond the abstract line-break artifact and a general objection to Fig. 4. The acknowledgement then turns this into a broader conclusion that the presentation is below the ICML bar, without identifying additional specific problems. We would therefore respectfully suggest separating fixable polish issues from the question of whether the technical claims and empirical evidence are sufficiently supported in the current submission.
> > >
> > > **W7.** We agree that the paper is focused on masked diffusion rather than on generative models in full generality. Our point is simply that this should be understood as the intended scope of the contribution, not as a flaw in itself. The paper targets a central bottleneck of masked diffusion, namely efficient revision of earlier mistakes, and evaluates the proposed approach on language modeling and code generation. We therefore believe this is best viewed as a focused contribution within an active model class.
> > >
> > > Our narrower concern is that several of the remaining criticisms still characterize the submission in a way that does not fully match the experiments and comparisons already present in the manuscript. We would therefore respectfully ask you to reconsider your current assessment in light of the evidence in the paper and the clarifications provided in the rebuttal.

---

### Official Review · Reviewer_ftoa · 2026-03-10

**Soundness:** 3
**Presentation:** 3
**Significance:** 3
**Originality:** 3
**Overall Recommendation:** 4
**Confidence:** 2

**Summary:**

This paper proposes a Guided Star-Shaped Sampler for masked diffusion. It formulates the sampling process under a star-shaped paradigm and introduces a learnable remasking module to identify and correct potential errors during generation. The sampler is compatible with standard pretrained masked diffusion models. Experimental results demonstrate the effectiveness and superiority of the proposed method.

**Compliance With Llm Reviewing Policy:**

Affirmed.

**Final Justification:**

The response has resolved my concerns, and I will keep my positive score unchanged.

**Key Questions For Authors:**

Please refer to the weaknesses

**Limitations:**

Yes

**Strengths And Weaknesses:**

Strengths

1. The Introduction clearly describes the limitations of existing methods and clearly presents the motivation for the Guided Star-Shaped Sample.

2. This paper provides many figures and tables to demonstrate the effectiveness of the proposed method.

Weaknesses

1. It is difficult to understand the working mechanism of the Guided Star-Shaped Sampler based solely on textual description; adding a methodological figure in the main paper would make it easier to understand.

2. Including some visualizations of the generated results in the main paper would help better demonstrate the superiority of the proposed method.
Question

---

> ### Author Rebuttal · Authors · 2026-03-27
>
> Thank you for your positive feedback and helpful suggestions on the presentation of our paper. We agree that the paper would benefit from a clearer visual summary in the main text.
>
> To address your specific points, we actually have these illustrations in the current submission and will make sure to bring them to the main part in our revision:
>
> - Regarding the methodological figure, we already include it in the appendix as Figure 14, which compares the sampling trajectories of MDLM, Star, and G-Star and illustrates the role of guided remasking. Due to space constraints in the submission version, this figure was placed in the appendix. In the revised version, we will move an adapted version of this figure into the main paper to make mechanism easier to follow.
>
> - Regarding qualitative visualizations, we also already provide them in the appendix. Appendix F was included specifically to visualize the refinement process and compare remasking behavior across samplers; in particular, it shows that G-Star performs structured, clustered revisions rather than indiscriminate remasking. In addition, Figure 13 presents a concrete text-generation example showing the refinement trajectory of G-Star and unguided Star+ from the same draft.
>
> Thank you again for these constructive suggestions.

---

> > ### Author Rebuttal · Reviewer_ftoa · 2026-04-03
> >
> > Thanks for the response. I have no further concerns, and I would like to keep my positive score.

---

### Official Review · Reviewer_gvuk · 2026-03-12

**Soundness:** 3
**Presentation:** 2
**Significance:** 3
**Originality:** 3
**Overall Recommendation:** 4
**Confidence:** 3

**Summary:**

This paper studies a limitation of standard masked diffusion: once a token is predicted, it cannot be corrected later. This is especially restrictive when using a small number of generation steps. To address this, the paper proposes a star-shaped forward process that allows previously generated tokens to be masked again and revised. The paper also introduces a lightweight error predictor to decide which tokens are more likely to be wrong, so that re-masking can be done more efficiently.

**Compliance With Llm Reviewing Policy:**

Affirmed.

**Final Justification:**

Thanks for authors' reply. My concerns have been resolved, so I will maintain my score.

**Key Questions For Authors:**

- In proposed method, the author treats every case where $\hat{x} _ {0,i} \neq x _ {0,i}$ as an error, but in open-ended generation, many tokens can be differ from the reference without actually being wrong. So is the predictor really learning the true errors, or just deviations from the reference?
- The predictor is trained on states obtained from corrupting ground-truth sequences, but at test time it is applied on trajectories generated by the model itself. Whether this distribution shift hurts calibration or makes the predictor less reliable during actual sampling?

**Limitations:**

yes

**Strengths And Weaknesses:**

Strength:
- The paper identifies the limitation of masked diffusion: once token predictions are made, they cannot be revised. It proposes a well-motivated re-masking solution to address this issue.
- The error predictor is lightweight. Experiments show that this head-only design achieves performance comparable to full fine-tuning.
- The star-shaped formulation naturally enables a revisable generation process while preserving a weighted cross-entropy training objective in the same form as standard masked diffusion. This could balancing interpretability and compatibility.

Weaknesses:
- The predictor is trained with $y _ i = \mathbf{1} [ \hat{x} _ {0,i} \neq x _ {0,i}]$  (line 170-176), which treats any deviation from the reference token as an error. In open-ended text generation this conflates true errors with valid alternative tokens, effectively encouraging reference imitation rather than detecting harmful tokens.
- The remasking decision is made based only on $\hat{x}_0$ (ine 173-177), ignoring the current timestep $t$, noisy state $x_t$, or model confidence. This makes the error detection partially observable and may lead to inconsistent remasking decisions.
- The predictor is trained on states derived from corrupted ground-truth sequences (lines 170–173) but applied during sampling on model-generated trajectories (lines 275–284), which may cause distribution shift and miscalibration.

Presentation Weakness(suggestion only, not affect the score):
- The method section is not very easy to follow. There is a lot of mathematical and textual description, but the high-level idea is less clear than it could be. A simple pipeline figure showing the overall framework would help a lot.

---

> ### Author Rebuttal · Authors · 2026-03-30
>
> We thank the reviewer for the careful, constructive feedback and the positive assessment of the motivation, lightweight error predictor, and empirical analysis.
>
> ### 1. Labels based on $\mathbf{1}[\hat{x}_0 \neq x_0]$
>
> We agree that this target does not perfectly track semantic errors. At the same time, we believe it is a meaningful training signal rather than merely a convenient approximation. To support this intuition, we tested whether a model trained with our current target treats semantically close words differently:
>
> **(a) Synonym sanity check.**
> We replaced a semantically valid token with a synonym (“great” $\rightarrow$ “good”, “large” $\rightarrow$ “big”, “beautiful” $\rightarrow$ “lovely”, and, in “The company released a new product last month”, “month” $\rightarrow$ “week”, yielding “The company released a new product last week”) and evaluated the remasker on both versions. The predictor assigned very low error probability to both tokens (e.g. 0.001 for “large”, 0.000 for “big”), suggesting that it is not simply flagging lexical deviation or imitating the reference.
>
> Additionally, we checked whether using other semantically informed targets improves the quality of the model compared to our approach.
>
> **(b) Random-corruption supervision.**
> We trained an alternative predictor on $x_0 \rightarrow \text{uniform corruption} \rightarrow x_0^{corr}$, so the target explicitly marks injected mistakes rather than deviations from a sampled draft. This gave **PPL = 14.82** and **DIV = 32.78** for 128-step generation, which is dominated by the original Pareto frontier.
>
> **(c) AR/NLL-based supervision.**
> We used GPT-2 Large to score tokens in $\hat{x}_0$ and labeled high-NLL tokens as errors after masking 10% of positions. This gave **PPL = 13.72** and **DIV = 27.06**, again inside the original Pareto frontier.
>
> Taken together, these results suggest that while $\mathbf{1}[\hat{x}_0 \neq x_0]$ is not a gold semantic correctness signal, it is a meaningful and practically useful proxy for refinement. We will revise the paper to make this point more clearly.
>
> ### 2. Why only condition on $\hat{x}_0$?
>
> In the standard MDLM training distribution, unmasked tokens in $x_t$ are copied from ground-truth $x_0$. Conditioning on $x_t$ therefore creates a strong shortcut: the predictor learns that tokens already visible in $x_t$ are correct and should not be remasked. Conditioning on model confidence creates the same shortcut in practice, pushing the remasker toward a conservative “do not revise” policy rather than learning to identify tokens that should be changed.
>
> For this reason, conditioning on $x_t$ or confidence risks collapsing the remasker into a conservative “do not revise” module. Restricting $g_\phi$ to the sampled draft $\hat{x}_0$ removes this shortcut and forces the predictor to judge correctness from the draft itself. We agree this motivation should be explained more clearly, and we will make this design choice more explicit in the revision.
>
> ### 3. Training/inference mismatch
>
> We agree this is an important point. More broadly, this kind of train-inference mismatch is a well-known issue in diffusion models and an important direction for future research. We therefore conducted an additional experiment to reduce this mismatch.
>
> For each clean sample $x_0$, we: (1) apply the standard noising process to obtain $x_t$; (2) denoise once to obtain $\hat{x}_0$; (3) re-noise $\hat{x}_0$ using the error predictor to obtain $x_t^{pred}$; (4) denoise again to obtain $\hat{x}_0^{pred}$; and (5) train the error predictor on $\mathbf{1}[\hat{x}_0^{pred} \neq x_0]$.
>
> This exposes the predictor to imperfect model-generated drafts during training, rather than only one-step denoised ground-truth corruptions.
>
> | Model | MAUVE | PPL | DIV |
> |---|---:|---:|---:|
> | G-Star-loop, 128 | 57.3 | 17.2 | 35.4 |
> | + reduced mismatch | 59.7 | 17.0 | 34.9 |
> | G-Star-loop, 256 | 60.9 | 12.7 | 30.9 |
> | + reduced mismatch | 61.7 | 12.7 | 30.7 |
> | G-Star-loop, 512 | 58.6 | 9.9 | 26.4 |
> | + reduced mismatch | 60.6 | 9.8 | 26.2 |
>
> The effect is modest: MAUVE improves slightly, while PPL and DIV are almost unchanged. However, training becomes roughly **2× slower**, since each update now requires additional denoising and remasking passes.
>
> Our takeaway is that the mismatch is real and theoretically meaningful, but in our setting it does not appear to be the dominant factor behind performance. We will highlight it more clearly as a future research direction in the revision.
>
> ### 4. Presentation and clarity
>
> We agree the high-level story can be clearer. In the camera-ready version, we will add a pipeline figure and streamline the method section to emphasize the two-phase generation story before the mathematical details.
>
> ---
>
> Overall, we appreciate the reviewer’s time and feedback. We hope these clarifications resolve the main concerns, and we would be grateful if the reviewer would reconsider the assessment in light of the points above.

---

> > ### Author Rebuttal · Reviewer_gvuk · 2026-04-03
> >
> > Thanks for authors' reply. My concerns have been resolved, so I will maintain my score.

---

### Official Review · Reviewer_ZDSa · 2026-03-16

**Soundness:** 3
**Presentation:** 3
**Significance:** 3
**Originality:** 3
**Overall Recommendation:** 4
**Confidence:** 4

**Summary:**

The authors propose Guided Star-Shaped Masked Diffusion (G-Star), a sampling algorithm that allows for iterative error correction during generation. Recognizing that standard remasking is inefficient, they introduce a "star-shaped" paradigm that conditionally decouples the next state from the current one by routing predictions through a fully denoised estimate. To make this efficient, they freeze the diffusion backbone and train a lightweight error-prediction head to explicitly target and remask only the most likely mistakes.

**Compliance With Llm Reviewing Policy:**

Affirmed.

**Final Justification:**

The rebuttal resolves the concerns. I misread Table 2. Qwen is the evaluator, not the generator. The rationale for isolating the sampling strategy in the code experiments is valid. Comparing against different foundational models introduces confounding variables since the method is an add on module. I am raising my score to a weak accept.

**Key Questions For Authors:**

Why were dedicated discrete diffusion models, particularly those optimized for code like DiffuCoder, Dream-coder or LLaDA, omitted from the code generation comparisons in Table 2?

The performance gains on Dream-Instruct 7B (Table 1) are relatively modest for coding tasks (+1.2 on HumanEval, +0.6 on MBPP). Does the added wall-clock latency of the G-Star refinement loop justify these marginal gains in a real-world deployment compared to simply scaling up the autoregressive baseline?

The Conala benchmark used in Table 2 is relatively small and old. How does the Qwen2.5B-Coder + G-Star setup perform on more modern, rigorous benchmarks like HumanEval or MBPP compared to standard discrete diffusion baselines?

**Limitations:**

While the authors provide a transparent breakdown of wall-clock latency in Appendix D showing that G-Star is faster than DDPD, they kind of gloss over the fact that it is still significantly slower than standard autoregressive generation with KV caching. The method requires multiple full forward passes of the backbone to do its targeted remasking, which inherently limits its utility in latency-sensitive applications.

**Strengths And Weaknesses:**

The theoretical motivation for why they built this is strong. The authors correctly point out a major flaw in confidence-based remasking: masked diffusion models are only supervised on masked tokens during training, meaning their confidence scores on already-generated tokens are uncalibrated and terrible for error detection. Training a tiny, parameter-efficient classification head to act as a targeted corrector is a highly practical, plug-and-play solution.

However, the code generation experiments in Table 2 are weak. They evaluate on the Conala benchmark using Qwen2.5B-Coder, but they only compare against MDLM and ReMDM. They explicitly acknowledge models like DiffuCoder and LLaDA in their related works section, yet fail to benchmark against them.

Furthermore, even in Table 1 where they evaluate Dream-Instruct 7B on HumanEval and MBPP, they are only proving that G-Star improves the base Dream model (e.g., pushing HumanEval from 53.7 to 54.9). They don't establish if this combined system is actually competitive with top-tier discrete diffusion models or standard autoregressive baselines.

---

> ### Author Rebuttal · Authors · 2026-03-27
>
> Thank you for the thoughtful review and for highlighting the practical value of targeted remasking. We believe that several concerns about the code experiments arise from a misunderstanding of the scope of our contribution. We clarify this below.
>
> Our paper does not propose a new standalone code generation model. G-Star is a plug-and-play sampler for existing masked diffusion models. In the paper, we first evaluate this sampler on open-ended text generation through the Pareto-front experiments. The code results then serve two follow-up purposes: **Table 2 tests whether the same sampler improvement transfers to code generation in a controlled same-backbone setting, while Table 1 tests whether it also transfers to a much stronger pretrained diffusion LLM.**
>
> With this in mind, **Table 2** is meant to isolate the effect of the **sampler** on code generation, not to compare all code models in the literature. For that reason, we keep the conditional diffusion backbone fixed and compare different sampling strategies on the same model: standard MDLM, ReMDM, and G-Star. This lets us measure whether **targeted remasking** improves the same generator in a new domain. In contrast, models such as DiffuCoder or LLaDA differ in model size, data, and training setup, so including them in Table 2 would mix the effect of the sampler with many other factors.
>
> **Table 1** addresses a different question: does G-Star still help when added to a much stronger pretrained diffusion LLM? This is why we evaluate it on Dream-Instruct 7B across seven benchmarks, including HumanEval and MBPP. The correct baseline here is **Dream-Instruct (Reproduced)**, since this is the matched setting. Under this comparison, G-Star improves HumanEval from **53.7 to 54.9** and MBPP from **58.0 to 59.4**, while also improving the other five benchmarks. The goal of this experiment is not to claim a new best code model, but to show that the same lightweight refinement idea remains effective at 7B scale.
>
> We also want to clarify one important point: **Qwen2.5B-Coder in Table 2 is not the generator.** It is only the **frozen evaluator** used to compute conditional perplexity. G-Star is applied to the conditional MDLM in Table 2.
>
> We agree that Conala is smaller and older than HumanEval or MBPP. We use it in Table 2 because it provides a clean conditional code generation setup where we can train a matched MDLM backbone and study the sampler in isolation. In other words, after establishing the effect of G-Star on text generation, Table 2 is meant to show that the same remasking strategy also remains useful when transferred to code generation. The more modern coding benchmarks already appear in Table 1 through Dream-Instruct 7B.
>
> On latency, we agree that autoregressive decoding with KV caching is faster, and we explicitly state this as a limitation. However, the central question of this paper is not whether masked diffusion can outperform autoregressive serving, but whether targeted remasking improves generation quality within masked diffusion models.
>
> We will revise the paper to make the progression of this analysis clearer: the Pareto-front results establish the effect of G-Star on text generation, **Table 2 extends this analysis to code generation in a controlled same-backbone setting, and Table 1 shows that the same refinement strategy also transfers to a strong 7B diffusion model.**
>
> We would be grateful if you would reconsider the score if these clarifications address your concerns.

---

> > ### Author Rebuttal · Reviewer_ZDSa · 2026-04-04
> >
> > Thanks for clarifying. The rebuttal resolves the concerns. I misread Table 2. Qwen is the evaluator, not the generator. The rationale for isolating the sampling strategy in the code experiments is also valid. Comparing against different foundational models introduces confounding variables since the method is an add on module. I am raising my score to a weak accept.

---

### Decision · Program_Chairs · 2026-04-30

**Decision:**

Accept (regular)

**Comment:**

This paper introduces Guided Star-Shaped Masked Diffusion, a plug-and-play module that uses a star-shaped paradigm for efficient error correction in pre-trained masked diffusion models. Final scores improved to three Weak Accepts and one Reject, with reviewers championing the paper post-rebuttal. The consensus for acceptance highlighted the method's strong theoretical motivation, its lightweight error-prediction head that elegantly addresses the flaws of confidence-based remasking, and the authors' successful rebuttal resolving core evaluation concerns.

For the camera-ready version, authors must incorporate their rebuttal commitments. Visually, they must add a pipeline figure to the main text (adapting Figure 14) and integrate qualitative trajectories (e.g., Figure 13, Appendix F) to better illustrate the method's mechanics. Textually, the manuscript must clarify Qwen's role strictly as an evaluator in Table 2, justify the proxy training target, and explain the conditioning choices used to prevent conservative "do not revise" shortcuts. Finally, authors should highlight the train-inference mismatch as future work and correct the abstract's line-break artifact.
To sum up, the final AC decision is **accept** .